# The probability flow ODE is provably fast

**Sitan Chen**[*]    **Sinho Chewi**[†]    **Holden Lee**[‡]    **Yuanzhi Li**[§]    **Jianfeng Lu**[¶]    **Adil Salim**[||]

## Abstract

We provide the first polynomial-time convergence guarantees for the probability flow ODE implementation (together with a corrector step) of score-based generative modeling with an OU forward process. Our analysis is carried out in the wake of recent results obtaining such guarantees for the SDE-based implementation (*i.e.*, denoising diffusion probabilistic modeling or DDPM), but requires the development of novel techniques for studying deterministic dynamics without contractivity. Through the use of a specially chosen corrector step based on the underdamped Langevin diffusion, we obtain better dimension dependence than prior works on DDPM ($O(\sqrt{d})$ vs. $O(d)$, assuming smoothness of the data distribution), highlighting potential advantages of the ODE framework.

## 1    Introduction

Score-based generative models (SGMs) [Soh+15; SE19; HJA20; DN21; Son+21a; Son+21b; VKK21] are a class of generative models which includes prominent image generation systems such as DALL·E 2 [Ram+22]. Their startling empirical success at data generation across a range of application domains has made them a central focus of study in the literature on deep learning [Aus+21; DN21; Kin+21; Shi+21; CSY22; Gna+22; Rom+22; Son+22; BV23; WHZ23]. In this paper, we aim to provide theoretical grounding for such models and thereby elucidate the mechanisms driving their remarkable performance.

Our work follows in the wake of numerous recent works which have provided convergence guarantees for denoising diffusion probabilistic models (DDPMs) [De +21; BMR22; De 22; LLT22; Liu+22; Pid22; WY22; Che+23a; CLL23; LLT23] and denoising diffusion implicit models (DDIMs) [CDD23]. We briefly recall that the generating process for SGMs is the time reversal of a certain diffusion process, and that DDPMs hinge upon implementing the reverse diffusion process as a stochastic differential equation (SDE) whose coefficients are learned via neural network training and the statistical technique of score matching [Hyv05; Vin11] (more detailed background is provided in §2). Among these prior works, the concurrent results of [Che+23a; LLT23] are remarkable because they require minimal assumptions on the data distribution (in particular, they do not assume log-concavity or similarly restrictive conditions) and they hold when the errors incurred during score matching are only bounded in an $L^2$ sense, which is both natural in view of the derivation of score matching (see [Hyv05; Vin11]) and far more realistic.[7] Subsequently, the work of [CLL23] significantly sharpened the analysis in the case when no smoothness assumptions are imposed on the data distribution.

---

[*]Harvard University, `sitan@seas.harvard.edu`

[†]Institute for Advanced Study, `schewi@ias.edu`

[‡]Johns Hopkins University, `hlee283@jhu.edu`

[§]Microsoft Research, `yuanzhili@microsoft.com`

[¶]Duke University, `jianfeng@math.duke.edu`

[||]Microsoft Research, `adilsalim@microsoft.com`

[7]It is unreasonable, for instance, to assume that the score errors are bounded in an $L^\infty$ sense, since we cannot hope to learn the score function in regions of the state space which are not well-covered by the training data.

37th Conference on Neural Information Processing Systems (NeurIPS 2023).

Taken together, these works paint an encouraging picture of our understanding of DDPMs which takes into account both the diversity of data in applications (including data distributions which are highly multimodal or supported on lower-dimensional manifolds), as well as the non-convex training process which is not guaranteed to accurately learn the score function uniformly in space.

Besides DDPMs, instead of implementing the time reversed diffusion as an SDE, it is also possible to implement it as an ordinary differential equation (ODE), called the *probability flow ODE* [Son+21b]; see §2. The ODE implementation is often claimed to be faster than the SDE implementation [Lu+22; ZC23], with the rationale being that ODE discretization is typically more accurate than SDE discretization, so that one could use a larger step size. Indeed, the discretization error usually depends on the regularity of the trajectories, which is $\mathcal{C}^1$ for ODEs but only $\mathcal{C}^{\frac{1}{2}-}$ for SDEs (*i.e.*, Hölder continuous with any exponent less than $\frac{1}{2}$) due to the roughness of the Brownian motion driving the evolution.

Far from being able to capture this intuition, current analyses of SGMs cannot even provide a *polynomial-time* analysis of the probability flow ODE. The key issue is that under our minimal assumptions (*i.e.*, without log-concavity of the data distribution), the underlying dynamics of either the ODE or SDE implementation are not contractive, and hence small errors quickly accumulate and are magnified. The aforementioned analyses of DDPMs managed to overcome this challenge by leveraging techniques specific to the analysis of SDEs, through which we now understand that *stochasticity* plays an important role in alleviating error accumulation. It is unknown, however, how to carry out the analysis for the purely deterministic dynamics inherent to the probability flow ODE.

Our first main contribution is to give the first convergence guarantees for SGMs with OU forward dynamics in which steps of the discretized probability flow ODE—referred to as *predictor steps*—are interleaved with *corrector steps* which runs the overdamped Langevin diffusion with estimated score, as pioneered in [Son+21b]. Our results are akin to prior works on DDPMs in that they hold under minimal assumptions on the data distribution and under $L^2$ bounds on the score estimation error, and our guarantees scale polynomially in all relevant problem parameters. Here, the corrector steps inject stochasticity which is crucial for our proofs; however, we emphasize that the use of corrector steps does *not* simply reduce the problem to applying existing DDPM analyses. Instead, we must develop an entirely new framework based on Wasserstein–to–TV regularization, which is of independent interest; see §4 for a detailed overview of our techniques. Our results naturally raise the question of whether the corrector steps are necessary in practice, and we discuss this further in §5.

When the data distribution is log-smooth, then the dimension dependence of prior results on DDPMs, as well as our first result for the probability flow ODE with overdamped corrector, both scale as $O(d)$. Does this contradict the intuition that ODE discretization is more accurate than SDE discretization? The answer is *no*; upon inspecting our proof, we see that the discretization error of the probability flow ODE is indeed smaller than what is incurred by DDPMs, and in fact allows for a larger step size of order $1/\sqrt{d}$. The bottleneck in our result stems from the use of the overdamped Langevin diffusion for the corrector steps. Taking inspiration from the literature on log-concave sampling (see, *e.g.*, [Che22] for an exposition), our second main contribution is to propose corrector steps based on the *underdamped* Langevin diffusion (see §2) which is known to improve the dimension dependence of sampling. In particular, we show that the probability flow ODE with underdamped Langevin corrector attains $O(\sqrt{d})$ dimension dependence. This dependence is better than what was obtained for DDPMs in [Che+23a; CLL23; LLT23] and therefore highlights the potential benefits of the ODE framework. We note that the benefit to which we refer is at *generation time*, and not at training time.

Previously, [JP22] have proposed a "noise–denoise" sampler using the underdamped Langevin diffusion, but to our knowledge, our work is the first to use it in conjunction with the probability flow ODE. Although we provide preliminary numerical experiments in the Appendix, we leave it as a question for future work to determine whether the theoretical benefits of the underdamped Langevin corrector are also borne out in practice.

## 1.1 Our contributions

In summary, our contributions are the following.

- We provide the first convergence guarantees for the probability flow ODE with overdamped Langevin corrector (DPOM; Algorithm 1).

- We propose an algorithm based on the probability flow ODE with underdamped Langevin corrector (DPUM; Algorithm 2).

- We provide the first convergence guarantees for DPUM. These convergence guarantees show improvement over (i) the complexity of DPOM ($O(\sqrt{d})$ vs $O(d)$) and (ii) the complexity of DDPMs, *i.e.*, SDE implementations of score-based generative models (again, $O(\sqrt{d})$ vs $O(d)$).

- We provide preliminary numerical experiments in a toy example showing that DPUM can sample from a highly non log-concave distribution (see Appendix). The numerical experiments are not among our main contributions and are provided for illustration only. The Python code can be found in the Supplementary material.

Our main theorem can be summarized informally as follows; see §3 for more detailed statements.

**Theorem 1** (Informal). *Assume that the score function along the forward process is $L$-Lipschitz, and that the data distribution has finite second moment. Assume that we have access to $\widetilde{O}(\varepsilon/\sqrt{L})$ $L^2$-accurate score estimates. Then, the probability flow ODE implementation of the reversed Ornstein–Uhlenbeck process, when interspersed with either the overdamped Langevin corrector (DPOM; Algorithm 1) or with the underdamped Langevin corrector (DPUM; Algorithm 2), outputs a sample whose law is $\varepsilon$-close in total variation distance to the data distribution, using $\widetilde{O}(L^3 d/\varepsilon^2)$ or $\widetilde{O}(L^2\sqrt{d}/\varepsilon)$ iterations respectively.*

Our result provides the *first* polynomial-time guarantees for the probability flow ODE implementation of SGMs, so long as it is combined with the use of corrector steps. Moreover, when the corrector steps are based on the underdamped Langevin diffusion, then the dimension dependence of our result is significantly smaller ($O(\sqrt{d})$ vs. $O(d)$) than prior works on the complexity of DDPMs, and thus provides justification for the use of ODE discretization in practice, compared to SDEs.

Our main assumption on the data is that the score functions along the forward process are Lipschitz continuous, which allows for highly non-log-concave distributions, yet does not cover non-smooth distributions such as distributions supported on lower-dimensional manifolds. However, as shown in [Che+23a; CLL23; LLT23], we can also obtain polynomial-time guarantees without this smoothness assumption via early stopping (see Remark 1).

## 1.2   Related works

The idea of using a time-reversed diffusion for sampling has been fruitfully exploited in the log-concave sampling literature via the *proximal sampler* [TP18; LST21; CE22; LC22; FYC23; LC23], as put forth in [Che+22], as well as through algorithmic stochastic localization [EMS22; MW23]. Although we do not aim to be comprehensive in our discussion of the literature, we mention, e.g., [ABV23; Che+23b] for alternative approaches for diffusion models. We also note that the recent work of [CDD23] obtained a discretization analysis for the probability flow ODE (without corrector) in KL divergence, though their bounds have a large dependence on $d$ and are exponential in the Lipschitz constant of the score integrated over time.

Since the original arXiv submission of this paper, there have been further works studying the probability flow ODE. The work of [BDD23] also studied the probability flow ODE, but without providing discretization guarantees (and with possibly exponential dependencies). The work [Li+23] provides polynomial-time guarantees for the probability flow ODE (without corrector steps), at the cost of larger polynomial dependencies and more stringent score assumptions (namely, bounds on the Jacobian of the score). Also, [PMM23] study another variant of the predictor-corrector framework.

## 2   Preliminaries

### 2.1   Score-based generative modeling

Let $q_\star$ denote the data distribution, *i.e.*, the distribution from which we wish to sample. In score-based generative modeling, we define a forward process $(q_t^{\rightarrow})_{t\geq 0}$ with $q_0^{\rightarrow} = q_\star$, which transforms our data distribution into noise. In this paper, we focus on the canonical choice of the Ornstein–Uhlenbeck

(OU) process,

$$\mathrm{d}x_t^\rightarrow = -x_t^\rightarrow \, \mathrm{d}t + \sqrt{2} \, \mathrm{d}B_t \,, \qquad x_0^\rightarrow \sim q_\star \,, \qquad q_t^\rightarrow := \mathrm{law}(x_t^\rightarrow) \,, \tag{1}$$

where $(B_t)_{t \geq 0}$ is a standard Brownian motion in $\mathbb{R}^d$. It is well-known that the OU process mixes rapidly (exponentially fast) to its stationary distribution, the standard Gaussian distribution $\gamma^d$.

Once we fix a time horizon $T > 0$, the time reversal of the SDE defined in (1) over $[0, T]$ is given by

$$\mathrm{d}x_t^\leftarrow = (x_t^\leftarrow + 2 \nabla \ln q_t^\leftarrow(x_t^\leftarrow)) \, \mathrm{d}t + \sqrt{2} \, \mathrm{d}B_t \,, \tag{2}$$

where $q_t^\leftarrow := q_{T-t}^\rightarrow$, and the reverse SDE is a generative model: when initialized at $x_0^\leftarrow \sim q_0^\leftarrow$, then $x_T^\leftarrow \sim q$. Since $q_0^\leftarrow = q_T^\rightarrow \approx \gamma^d$, the reverse SDE transforms samples from $\gamma^d$ (i.e., pure noise) into approximate samples from $q_\star$. In order to implement the reverse SDE, however, one needs to estimate the score functions $\nabla \ln q_t^\leftarrow$ for $t \in [0, T]$ using the technique of score matching [Hyv05; Vin11]. In practice, the score estimates are produced via a deep neural network, and our main assumption is that these score estimates are accurate in an $L^2$ sense (see Assumption 4). This gives rise to the denoising diffusion probabilistic modeling (DDPM) algorithm.

**Notation.** Since the reverse process is the primary object of interest, we drop the arrow $\leftarrow$ from the notation for simplicity; thus, $q_t := q_t^\leftarrow$. We will always denote the forward process with the arrow $\rightarrow$.

For each $t \in [0, T]$, let $s_t$ denote the estimate for the score $\nabla \ln q_t = \nabla \ln q_t^\leftarrow$.

## 2.2 Probability flow ODE (predictor steps)

Instead of running the reverse SDE (2), there is in fact an alternative process $(x_t)_{t \in [0, T]}$ which evolves according to an ODE (and hence evolves deterministically), and yet has the same marginals as (2). This alternative process, called the *probability flow ODE*, can also be used for generative modeling.

One particularly illuminating way of deriving the probability flow ODE is to invoke the celebrated theorem, due to [JKO98], that the OU process is the Wasserstein gradient flow of the KL divergence functional (i.e. relative entropy) $\mathsf{KL}(\cdot \parallel \gamma^d)$. From the general theory of Wasserstein gradient flows (see [AGS08; San15]), the Wasserstein gradient flow $(\mu_t)_{t \geq 0}$ of a functional $\mathcal{F}$ can be implemented via the dynamics

$$\dot{z}_t = -[\nabla_{W_2} \mathcal{F}(\mu_t)](z_t) \,, \qquad z_0 \sim \mu_0 \,,$$

in that $z_t \sim \mu_t$ for all $t \geq 0$. Applying this to $\mathcal{F} := \mathsf{KL}(\cdot \parallel \gamma^d)$, we arrive at the forward process

$$\dot{x}_t^\rightarrow = -\nabla \ln \Big( \frac{q_t^\rightarrow}{\gamma^d} \Big)(x_t^\rightarrow) = -x_t^\rightarrow - \nabla \ln q_t^\rightarrow(x_t^\rightarrow) \,. \tag{3}$$

Setting $x_t := x_{T-t}^\rightarrow$, it is easily seen that the time reversal of (3) is

$$\dot{x}_t = x_t + \nabla \ln q_t(x_t) \,, \quad \text{i.e.,} \quad \dot{x}_t = x_t + \nabla \ln q_{T-t}^\rightarrow(x_t) \,, \tag{4}$$

which is called the probability flow ODE. In this paper, the interpretation of the probability flow ODE as a reverse Wasserstein gradient flow is only introduced for interpretability, and the reader who is unfamiliar with Wasserstein calculus can take (4) to be the definition of the probability flow ODE. Crucially, it has the property that if $x_0 \sim q_0$, then $x_t \sim q_t$ for all $t \in [0, T]$.

We can discretize the ODE (4). Fixing a step size $h > 0$, replacing the score function $\nabla \ln q_t$ with the estimated score given by $s_t$, and applying the exponential integrator to the ODE (i.e., exactly integrating the linear part), we arrive at the discretized process

$$x_{t+h} = x_t + \int_0^h x_{t+u} \, \mathrm{d}u + h \, s_t(x_t) = \exp(h) \, x_t + (\exp(h) - 1) \, s_t(x_t) \,. \tag{5}$$

## 2.3 Corrector steps

Let $q$ be a distribution over $\mathbb{R}^d$, and write $U$ as a shorthand for the potential $-\ln q$.

**Overdamped Langevin.** The *overdamped Langevin diffusion* with potential $U$ is a stochastic process $(x_t)_{t\geq 0}$ over $\mathbb{R}^d$ given by

$$\mathrm{d}x_t = -\nabla U(x_t)\,\mathrm{d}t + \sqrt{2}\,\mathrm{d}B_t\,.$$

The stationary distribution of this diffusion is $q \propto \exp(-U)$.

We also consider the following discretized process where $-\nabla U$ is replaced by a *score estimate* $s$. Fix a step size $h > 0$ and let $(\widehat{x}_t)_{t\geq 0}$ over $\mathbb{R}^d$ be given by

$$\mathrm{d}\widehat{x}_t = s(\widehat{x}_{\lfloor t/h\rfloor\, h})\,\mathrm{d}t + \sqrt{2}\,\mathrm{d}B_t\,.$$

**Underdamped Langevin.** Given a friction parameter $\gamma > 0$, the corresponding *underdamped Langevin diffusion* is a stochastic process $(z_t, v_t)_{t\geq 0}$ over $\mathbb{R}^d \times \mathbb{R}^d$ given by

$$\mathrm{d}z_t = v_t\,\mathrm{d}t\,,$$
$$\mathrm{d}v_t = -(\nabla U(z_t) + \gamma v_t)\,\mathrm{d}t + \sqrt{2\gamma}\,\mathrm{d}B_t\,.$$

The stationary distribution of this diffusion is $q \otimes \gamma^d$.

We also consider the following discretized process, where $-\nabla U$ is replaced by a score estimate $s$. Let $(\widehat{z}_t, \widehat{v}_t)_{t\geq 0}$ over $\mathbb{R}^d \times \mathbb{R}^d$ be given by

$$\mathrm{d}\widehat{z}_t = \widehat{v}_t\,\mathrm{d}t\,, \tag{6}$$
$$\mathrm{d}\widehat{v}_t = (s(\widehat{z}_{\lfloor t/h\rfloor\, h}) - \gamma\widehat{v}_t)\,\mathrm{d}t + \sqrt{2\gamma}\,\mathrm{d}B_t\,.$$

**Diffusions as corrector steps.** At time $t$, the law of the ideal reverse process (4) initialized at $q_0$ is $q_t$. However, errors are accumulated through the course of the algorithm: the error from initializing at $\gamma^d$ rather than at $q_0$; errors arising from discretization of (4); and errors in estimating the score function. That's why the law of the algorithm's iterate will not be exactly $q_t$. We propose to use either the overdamped or the underdamped Langevin diffusion with stationary distribution $q_t$ and estimated score as a corrector, in order to bring the law of the algorithm iterate closer to $q_t$. In the case of the underdamped Langevin diffusion, this is done by drawing an independent Gaussian random variable $\widehat{v}_0 \sim \gamma^d$, running the system (6) starting from $(\widehat{z}_0, \widehat{v}_0)$ (where $\widehat{z}_0$ is the current algorithm iterate) for some time $t$, and then keeping $\widehat{z}_t$. In our theoretical analysis, the use of corrector steps boosts the accuracy and efficiency of the SGM.

# 3 Results

## 3.1 Assumptions

We make the following mild assumptions on the data distribution $q_\star$ and on the score estimate $s$.

**Assumption 1** (second moment bound). *We assume that $\mathfrak{m}_2^2 := \mathbb{E}_{q_\star}[\|\cdot\|^2] < \infty$.*

**Assumption 2** (Lipschitz score). *For all $t \in [0, T]$, the score $\nabla \ln q_t$ is $L$-Lipschitz, for some $L \geq 1$.*

**Assumption 3** (Lipschitz score estimate). *For all $t$ for which we need to estimate the score function in our algorithms, the score estimate $s_t$ is $L$-Lipschitz.*

**Assumption 4** (score estimation error). *For all $t$ for which we need to estimate the score function in our algorithms,*

$$\mathbb{E}_{q_t}[\|s_t - \nabla \ln q_t\|^2] \leq \varepsilon_{\mathsf{sc}}^2\,.$$

Assumptions 1, 2, and 4 are standard and were shown in [Che+23a; CLL23; LLT23] to suffice for obtaining polynomial-time convergence guarantees for DDPMs. The new condition that we require in our analysis is Assumption 3, which was used in [LLT22] but ultimately shown to be unnecessary for DDPMs. We leave it as an open question whether this can be lifted in the ODE setting.

*Remark* 1. As observed in [Che+23a; CLL23; LLT23], Assumption 2 can be removed via early stopping, at the cost of polynomially larger iteration complexity. The idea is that if $q_\star$ has compact support but does not necessarily satisfy Assumption 2 (*e.g.*, if $q_\star$ is supported on a compact and lower-dimensional manifold), then $q_\delta^{\rightarrow}$ will satisfy Assumption 2 if $\delta > 0$. By applying our analysis up to time $T - \delta$ instead of time $T$, one can show that a suitable projection of the output distribution is close in Wasserstein distance to $q_\star$ (see [CLL23, Corollary 2.4] or [Che+23a, Corollary 5]). For brevity, we do not consider this extension of our results here.

## 3.2 Algorithms

We provide the pseudocode for the two algorithms we consider, *Diffusion Predictor + Overdamped Modeling* (DPOM) and *Diffusion Predictor + Underdamped Modeling* (DPUM), in Algorithms 1 and 2 respectively. The only difference between the two algorithms is in the corrector step, which we highlight in Algorithm 2. For simplicity, we take the total amount of time $T$ to be equal to $N_0/L + h_{\mathsf{pred}}$ for an integer $N_0 \geq 1$, and we assume that $1/L$ is a multiple of $h_{\mathsf{pred}}$ and that $h_{\mathsf{pred}}$ is a multiple of $\delta = \Theta\big(\frac{\varepsilon^2}{L^2 (d \vee \mathfrak{m}_2^2)}\big)$.

We consider two stages: in the first stage, which lasts until time $N_0/L = T - h_{\mathsf{pred}}$, we intersperse predictor epochs (run for time $1/L$, discretized with step size $h_{\mathsf{pred}}$) and corrector epochs (run for time $\Theta(1/L)$ for the overdamped corrector or for time $\Theta(1/\sqrt{L})$ for the underdamped corrector, and discretized with step size $h_{\mathsf{corr}}$). The second stage lasts from time $T - h_{\mathsf{pred}}$ to time $T - \delta$, and we incorporate geometrically decreasing step sizes for the predictor. Note that this implies that our algorithm uses *early stopping*.

---

**Algorithm 1:** DPOM$(T, h_{\mathsf{pred}}, h_{\mathsf{corr}}, s)$

**Input:** Total time $T$, predictor step size $h_{\mathsf{pred}}$, corrector step size $h_{\mathsf{corr}}$, score estimates $s$
**Output:** Approximate sample from the data distribution $q_\star$

1   Draw $\widehat{x}_0 \sim \gamma^d$.
2   **for** $n = 0, 1, \ldots, N_0 - 1$ **do**
3      **Predictor:** Starting from $\widehat{x}_{n/L}$, run the discretized probability flow ODE (5) from time $\frac{n}{L}$ to $\frac{n+1}{L}$ with step size $h_{\mathsf{pred}}$ and estimated scores to obtain $\widehat{x}'_{(n+1)/L}$.
4      **Corrector:** Starting from $\widehat{x}'_{(n+1)/L}$, run overdamped Langevin Monte Carlo for total time $\Theta(1/L)$ with step size $h_{\mathsf{corr}}$ and score estimate $s_{(n+1)/L}$ to obtain $\widehat{x}_{(n+1)/L}$.
5   **Predictor:** Starting from $\widehat{x}_{T - h_{\mathsf{pred}}}$, run the discretized probability flow ODE (5) with step sizes $h_{\mathsf{pred}}/2, h_{\mathsf{pred}}/4, h_{\mathsf{pred}}/8, \ldots, \delta$ and estimated scores to obtain $\widehat{x}'_{T-\delta}$.
6   **Corrector:** Starting from $\widehat{x}'_{T-\delta}$, run overdamped Langevin Monte Carlo for total time $\Theta(1/L)$ with step size $h_{\mathsf{corr}}$ and score estimate $s_{T-\delta}$ to obtain $\widehat{x}_{T-\delta}$.
7   **return** $\widehat{x}_{T-\delta}$

---

**Algorithm 2:** DPUM$(T, h_{\mathsf{pred}}, h_{\mathsf{corr}}, s)$

**Input:** Total time $T$, predictor step size $h_{\mathsf{pred}}$, corrector step size $h_{\mathsf{corr}}$, score estimates $s$
**Output:** Approximate sample from the data distribution $q_\star$

1   Draw $\widehat{x}_0 \sim \gamma^d$.
2   **for** $n = 0, 1, \ldots, N_0 - 1$ **do**
3      **Predictor:** Starting from $\widehat{x}_{n/L}$, run the discretized probability flow ODE (5) from time $\frac{n}{L}$ to $\frac{n+1}{L}$ with step size $h_{\mathsf{pred}}$ and estimated scores to obtain $\widehat{x}'_{(n+1)/L}$.
4      **Corrector:** Starting from $\widehat{x}'_{(n+1)/L}$, run underdamped Langevin Monte Carlo for total time $\Theta(1/\sqrt{L})$ with step size $h_{\mathsf{corr}}$ and score estimate $s_{(n+1)/L}$ to obtain $\widehat{x}_{(n+1)/L}$.
5   **Predictor:** Starting from $\widehat{x}_{T - h_{\mathsf{pred}}}$, run the discretized probability flow ODE (5) with step sizes $h_{\mathsf{pred}}/2, h_{\mathsf{pred}}/4, h_{\mathsf{pred}}/8, \ldots, \delta$ and estimated scores to obtain $\widehat{x}'_{T-\delta}$.
6   **Corrector:** Starting from $\widehat{x}'_{T-\delta}$, run underdamped Langevin Monte Carlo for total time $\Theta(1/\sqrt{L})$ with step size $h_{\mathsf{corr}}$ and score estimate $s_{T-\delta}$ to obtain $\widehat{x}_{T-\delta}$.
7   **return** $\widehat{x}_{T-\delta}$

---

## 3.3 Convergence guarantees

Our main results are the following convergence guarantees for the two predictor-corrector schemes described in §3.2:

**Theorem 2** (DPOM). *Suppose that Assumptions [1–4] hold. If $\widehat{q}$ denotes the output of DPOM (Algorithm [1]) with $\delta \asymp \frac{\varepsilon^2}{L^2\,(d \vee \mathfrak{m}_2^2)}$, then*

$$\mathsf{TV}(\widehat{q}, q_\star) \lesssim (\sqrt{d} \vee \mathfrak{m}_2)\exp(-T) + L^2 T d^{1/2} h_{\mathsf{pred}} + L^{3/2} T d^{1/2} h_{\mathsf{corr}}^{1/2} + L^{1/2} T \varepsilon_{\mathsf{sc}} + \varepsilon\,. \qquad (7)$$

*In particular, if we set $T = \Theta\big(\ln(\frac{d \vee \mathfrak{m}_2^2}{\varepsilon^2})\big)$, $h_{\mathsf{pred}} = \widetilde{\Theta}(\frac{\varepsilon}{L^2 d^{1/2}})$, $h_{\mathsf{corr}} = \widetilde{\Theta}(\frac{\varepsilon^2}{L^3 d})$, and if the score estimation error satisfies $\varepsilon_{\mathsf{sc}} \leq \widetilde{O}(\frac{\varepsilon}{\sqrt{L}})$, then we can obtain TV error $\varepsilon$ with a total iteration complexity of $\widetilde{\Theta}(\frac{L^3 d}{\varepsilon^2})$ steps.*

The five terms in the bound (7) correspond, respectively, to: the convergence of the forward (OU) process; the discretization error from the predictor steps; the discretization error from the corrector steps; the score estimation error; and the early stopping error.

Theorem [2] recovers nearly the same guarantees as the one in [Che+23a; CLL23; LLT23], but for the probability flow ODE with overdamped Langevin corrector instead of the reverse SDE without corrector. Recall also from Remark [1] that our results can easily be extended to compactly supported data distributions without smooth score functions. This covers essentially all distributions encountered in practice. Therefore, our result provides compelling theoretical justification complementing the empirical efficacy of the probability flow ODE, which was hitherto absent from the literature.

However, in Theorem [2], the iteration complexity is dominated by the corrector steps. Next, we show that by replacing the overdamped LMC with underdamped LMC, we can achieve a quadratic improvement in the number of steps, considering the dependence on $d$. As discussed in the Introduction, this highlights the potential benefits of the ODE framework over the SDE.

**Theorem 3** (DPUM). *Suppose that Assumptions [1–4] hold. If $\widehat{q}$ denotes the output of DPUM (Algorithm [2]) with $\delta \asymp \frac{\varepsilon^2}{L^2\,(d \vee \mathfrak{m}_2^2)}$, then*

$$\mathsf{TV}(\widehat{q}, q_\star) \lesssim (\sqrt{d} \vee \mathfrak{m}_2)\exp(-T) + L^2 T d^{1/2} h_{\mathsf{pred}} + L^{3/2} T d^{1/2} h_{\mathsf{corr}} + L^{1/2} T \varepsilon_{\mathsf{sc}} + \varepsilon\,.$$

*In particular, if we set $T = \Theta\big(\ln(\frac{d \vee \mathfrak{m}_2^2}{\varepsilon^2})\big)$, $h_{\mathsf{pred}} = \widetilde{\Theta}(\frac{\varepsilon}{L^2 d^{1/2}})$, $h_{\mathsf{corr}} = \widetilde{\Theta}(\frac{\varepsilon}{L^{3/2} d^{1/2}})$, and if the score estimation error satisfies $\varepsilon_{\mathsf{sc}} \leq \widetilde{O}(\frac{\varepsilon}{\sqrt{L}})$, then we can obtain TV error $\varepsilon$ with a total iteration complexity of $\widetilde{\Theta}(\frac{L^2 d^{1/2}}{\varepsilon})$ steps.*

## 4 Proof overview

Here we give a detailed technical overview for the proof of our main results, Theorems [2] and [3]. As in [Che+23a; CLL23; LLT23], the three sources of error that we need to keep track of are (1) estimation of the score function; (2) discretization of time when implementing the probability flow ODE and corrector steps; and (3) initialization of the algorithm at $\gamma^d$ instead of the true law of the end of the forward process, $q_0 = q_{\overrightarrow{T}}$. It turns out that (1) is not so difficult to manage as soon as we can control (2) and (3). Furthermore, as in prior work, we can easily control (3) via the data-processing inequality: the total variation distance between the output of the algorithm initialized at $q_0$ versus at $\gamma^d$ is at most $\mathsf{TV}(q_{\overrightarrow{T}}, \gamma^d)$, which is exponentially small in $T$ by rapid mixing of the OU process. So henceforth in this overview, let us assume that both the algorithm and the true process are initialized at $q_0$. It remains to control (2).

**Failure of existing approaches.** In the SDE implementation of diffusion models, prior works handled (2) by directly bounding a strictly larger quantity, namely the KL divergence between the laws of the *trajectories* of the algorithm and the true process; by Girsanov's theorem, this has a clean formulation as an integrated difference of drifts. Unfortunately, in the ODE implementation, this KL divergence is infinite: in the absence of stochasticity in the reverse process, these laws over trajectories are not even absolutely continuous with respect to each other.

In search of an alternative approach, one might try a Wasserstein analysis. As a first attempt, we could couple the initialization of both processes and look at how the distance between them changes over time. If $(\widehat{x}_t)_{0 \leq t \leq T}$ and $(x_t)_{0 \leq t \leq T}$ denote the algorithm and true process, then smoothness of the score function allows us to naïvely bound $\partial_t \mathbb{E}[\|\widehat{x}_t - x_t\|^2]$ by $O(L)\,\mathbb{E}[\|\widehat{x}_t - x_t\|^2]$. While this ensures that the processes are close if run for time $\ll 1/L$, it does not rule out the possibility that they drift apart exponentially quickly after time $1/L$.

**Restarting the coupling—first attempt.** What we would like is some way of "restarting" this coupling before the processes drift too far apart, to avoid this exponential compounding. We now motivate how to achieve this by giving an argument that is incorrect but nevertheless captures the intuition for our approach. Namely, let $p_t \coloneqq \text{law}(\widehat{x}_t)$ denote the law of the algorithm, let $P_{\mathsf{ODE}}^{t_0,h}$ denote the result of running the ideal probability flow ODE for time $h$ starting from time $t_0$, and let $\widehat{P}_{\mathsf{ODE}}^{t_0,h}$ denote the same but for the discretized probability flow ODE with estimated score. For $h \lesssim 1/L$, consider the law of the two processes at time $2h$, i.e.,

$$p_{2h} = q_0 \widehat{P}_{\mathsf{ODE}}^{0,2h} \qquad \text{and} \qquad q_{2h} = q_0 P_{\mathsf{ODE}}^{0,2h}. \tag{8}$$

The discussion above implies that $q_0 P_{\mathsf{ODE}}^{0,h}$ and $q_0 \widehat{P}_{\mathsf{ODE}}^{0,h}$ are close in 2-Wasserstein distance, so by the data-processing inequality, this implies that $q_0 P_{\mathsf{ODE}}^{0,h} \widehat{P}_{\mathsf{ODE}}^{h,h}$ and $q_0 \widehat{P}_{\mathsf{ODE}}^{0,h} \widehat{P}_{\mathsf{ODE}}^{h,h}$ are also close. To show that $p_{2h}$ and $q_{2h}$ in Eq. (8) are close, it thus suffices to show that $q_0 P_{\mathsf{ODE}}^{0,2h}$ and $q_0 P_{\mathsf{ODE}}^{0,h} \widehat{P}_{\mathsf{ODE}}^{h,h}$ are close. But these two distributions are given by running the algorithm and the true process for time $h$, both starting from $q_0 P_{\mathsf{ODE}}^{0,h}$. So if we "restart" the coupling by coupling the processes based on their locations at time $h$, rather than time $0$, of the reverse process, we can again apply the naïve Wasserstein analysis.

At this juncture, it would seem that we have miraculously sidestepped the exponential blowup and shown that the expected distance between the processes only increases linearly over time! The issue of course is in the application of the "data-processing inequality," which simply does not hold for the Wasserstein distance.

**Restarting the coupling with a corrector step.** This is where the corrector comes in. The idea is to use *short-time regularization*: if we apply a small amount of noise to two distributions which are already close in Wasserstein, then they become close in KL divergence, for which a data-processing inequality holds. The upshot is that if the noise doesn't change the distributions too much, then we can legitimately restart the coupling as above and prove that the distance between the processes, now defined by interleaving the probability flow ODE and its discretization with periodic injections of noise, increases only linearly in time.

It turns out that naïve injection of noise, e.g., convolution with a Gaussian of small variance, is somewhat wasteful as it fails to preserve the true process and leads to poor polynomial dependence in the dimension. On the other hand, if we instead run the overdamped Langevin diffusion with potential chosen so that the law of the true process is stationary, then we can recover the linear in $d$ dependence of Theorem 2. Then by replacing overdamped Langevin diffusion with its underdamped counterpart, which has the advantage of much smoother trajectories, we can obtain the desired quadratic speedup in dimension dependence in Theorem 3.

**Score perturbation lemma.** In addition to the switch from SDE to ODE and the use of the underdamped corrector, a third ingredient is essential to our improved dimension dependence. The former two ensure that the trajectory of our algorithm is smoother than that of DDPMs, so that even over time windows that scale with $1/\sqrt{d}$, the process does not change too much. By extension, as the score functions are Lipschitz, this means that any fixed score function evaluated over iterates in such a window does not change much. This amounts to controlling discretization error in *space*.

It is also necessary to control discretization error in *time*, i.e., proving what some prior works referred to as a *score perturbation lemma* [LLT22]. That is, for any fixed *iterate* $x$, we want to show that the score function $\nabla \ln q_t(x)$ does not change too much as $t$ varies over a small window. Unfortunately, prior works were only able to establish this over windows of length $1/d$. In this work, we improve this to windows of length $1/\sqrt{d}$ (see Lemma 3 and Corollary 1).

In our proof, we bound the squared $L^2$ norm of the derivative of the score along the trajectory of the ODE. The score function evaluated at $y$ can be expressed as $\mathbb{E}_{P_{0|t}(\cdot|y)}[\nabla U]$; here, the posterior distribution $P_{0|t}(\cdot \mid y)$ is essentially the prior $q_\star$ tilted by a Gaussian of variance $O(t)$. Hence we need to bound the change in the expectation when we change the distribution from $P_{0|t}$ to $P_{0|t+\Delta t}$; because $\nabla U$ is $L$-Lipschitz, we can bound this by the Wasserstein distance between the distributions. For small enough $t$, $P_{0|t}$ is strongly log-concave, and a transport cost inequality bounds this in terms of KL divergence, which is more easily bounded. Indeed, we can bound it with the KL divergence between the joint distributions $P_{0,t}$ and $P_{0,t+\Delta t}$, which reduces to bounding the KL divergence between Gaussians of unequal variance.

However, since our score perturbation lemma degrades near the beginning of the forward process, we require better control of the discretization error during this part of the algorithm, hence leading to our choice of geometrically decreasing step sizes. Alternatively, we could use a two-stage step size schedule, see Remark 4.

## 5    Conclusion

In this work, we have provided the first polynomial-time guarantees for the probability flow ODE implementation of SGMs with corrector steps and exhibited improved dimension dependence of the ODE framework over prior results for DDPMs (*i.e.,* the SDE framework). Our analysis raises questions relevant for practice, of which we list a few.

- Although we need the corrector steps for our proof, are they in fact necessary for the algorithm to work efficiently in practice?
- Is it possible to obtain even better dimension dependence, perhaps using higher-order solvers and stronger smoothness assumptions?
- Can we obtain improved dimension dependence even in the non-smooth setting, compared to the result of [CLL23]?

We also list several limitations of our work, namely:

- Our analysis only covers the probability flow ODE corresponding to the OU forward process. We leave the study of more general dynamics for future study.
- Our guarantees require the score function to be learned to $L^2$ accuracy $\widetilde{O}(\varepsilon/\sqrt{L})$, which is more stringent than the prior works [Che+23a; CLL23; LLT23] and may be an technical artefact of our proof.
- We have not validated our theoretical findings with large-scale experiments. In particular, it is still unclear whether flow-based methods can outperform the standard DDPM algorithm in practical, high dimensional settings.

## 6    Acknowledgments

We want to thank Yin Tat Lee for his valuable comments, shaping the direction of this research project in its early stages. SC was supported by NSF Award 2103300 for part of this work.

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

## A  Notation and overview

In this section, we collect together the notation used throughout the proofs and provide a road map for the end-to-end analysis in §E.

Throughout the analysis, Assumptions 1–4 are in full force.

We will reserve $q$ for the law of the reverse process (and denote the forward process by $q^{\rightarrow}$ when needed). In §E, the law of the algorithm is denoted by $p$.

We use the following Markov kernels:

1. $P_{\mathsf{ODE}}^{t,h}$ is the output of running the ODE for time $h$, starting at (reverse) time $t$.
2. $P_{\mathsf{LD}}$ (resp. $P_{\mathsf{ULD}}$) is the output of running the continuous-time overdamped (resp. under-damped) Langevin diffusion for time $h$. In this notation, we have suppressed mention of the stationary distributions of the diffusion, which will be provided by context.
3. $\widehat{P}_{\mathsf{ODE}}^{t,h}$ and $\widehat{P}_{\mathsf{LMC}}$ (resp. $\widehat{P}_{\mathsf{ULMC}}$) are the corresponding processes once discretized and using the estimated score.

For the ODE, we are more precise with the notation because even within a single epoch of predictor steps, the kernel for the probability flow ODE depends on time (as opposed to the kernels for the diffusions, which are constant within any epoch of corrector steps); moreover, for our analysis in §E, we also need to take time-varying step sizes for the predictor steps. We will omit the dependencies on $t$ and $h$ when clear from context. When $P = P_{\mathsf{ODE}}$ or $\widehat{P}_{\mathsf{ODE}}$, we use $P^{t,h_1,\dots,h_N}$ to denote $P^{t,h_1} P^{t+h_1,h_2} \cdots P^{t+h_1+\cdots+h_{N-1},h_N}$ (we compose kernels on the right).

We refer to §4 for a high-level description of the proof strategy. We begin in §B with our improved score perturbation lemma (Corollary 1); this is the only section of the analysis which is indexed by *forward* time (instead of reverse time). In Lemma 5 in §C, we establish our main result for the predictor steps, which combines together standard ODE discretization analysis with the score perturbation lemma of §B. Since Corollary 1 degrades near the end of the reverse process (or equivalently, near the start of the forward process, when the regularization has not yet kicked in), our analysis requires a geometrically decreasing step size schedule, which leads to the two-stage Algorithms 1 and 2.

In §D, we prove our main regularization results for the overdamped corrector (Theorem 4) and the underdamped corrector (Theorem 5). Finally, we put together the various constituent results in the end-to-end analysis in §E.

## B  Score perturbation

In this section, we prove a score perturbation lemma which refines that of [LLT22]. This improved lemma is necessary in order to obtain $O(\sqrt{d})$ dependence for the probability flow ODE.

**Lemma 1** (Score perturbation). *Suppose $p_t = p_0 * \mathcal{N}(0, tI)$ and $x_0 \sim p_0$, $\dot{x}_t = -\frac{1}{2} \nabla \ln p_t(x_t)$. Suppose that $\|\nabla^2 \ln p_{(t-\frac{1}{2L})\vee 0}(x)\|_{\mathsf{op}} \leq L$ for all $x$. Then*

$$\mathbb{E}[\|\partial_t \nabla \ln p_t(x_t)\|^2] \leq L^2 d \left( L + \frac{1}{t} \right).$$

*Proof.* Without loss of generality, we may assume $t \leq \frac{1}{2L}$, as otherwise, noting that $p_t = p_{t-\frac{1}{2L}} * N(0, \frac{1}{2L}I)$, we may replace $p_0$ with $p_{t-\frac{1}{2L}}$ and $t$ with $\frac{1}{2L}$. Suppose $p_0(x) = e^{-V(x)}$. Let $P_{0,t}$ denote the joint distribution of $(X_0, X_t)$ where $X_t = X_0 + \sqrt{t} Z$ with $Z \sim \mathcal{N}(0, I)$ independent of $X_0$, and let $P_{0|t}(\cdot \mid x_t)$ denote the conditional distribution of $X_0$ given $X_t = x_t$. We first note that since

$$\ln p_t(y) = \ln \int \exp\bigl(-V(x) - \frac{1}{2t} \|y - x\|^2\bigr) \, \mathrm{d}x \,,$$

we have the following calculations:

$$\nabla \ln p_t(y) = -\frac{1}{t} \mathbb{E}_{P_{0|t}(\cdot|y)}(y - \cdot) = -\mathbb{E}_{P_{0|t}(\cdot|y)}(\nabla V) \,,$$

$$\nabla^2 \ln p_t(y) = \mathrm{Cov}_{P_{0|t}(\cdot|y)}(\nabla V) - \mathbb{E}_{P_{0|t}(\cdot|y)}(\nabla^2 V).$$

Using $\dot{x}_t = -\frac{1}{2}\nabla \ln p_t(x_t)$, we calculate

$$\partial_t \nabla \ln p_t(x_t) = [\partial_t \nabla \ln p_t(y)]|_{y=x_t} - \frac{1}{2}\nabla^2 \ln p_t(x_t)\,\nabla \ln p_t(x_t). \tag{9}$$

We bound each term above separately. For the first term, a quick calculation shows that $\partial_t \nabla \ln p_t(y) = -\mathrm{Cov}_{P_{0|t}(\cdot|y)}\big(\frac{\|y-\cdot\|^2}{2t^2}, \nabla V\big)$ is finite a.s.: by Cauchy–Schwarz, it suffices to show $\mathbb{E}_{P_{0|t}(\cdot|y)}[\|y-\cdot\|^4]$ and $\mathbb{E}_{P_{0|t}(\cdot|y)}[\|\nabla V\|^2]$ are finite for all $y$, and this follows because for $t \leq \frac{1}{2L}$ the measure $P_{0|t}(\cdot \mid y)$ is strongly log-concave and $\|\nabla V\|^2$ can be bounded by a quadratic. Because $\nabla V$ is $L$-Lipschitz,

$$\begin{aligned}
\|[\partial_t \nabla \ln p_t(y)]|_{y=x_t}\|^2 &= \|[\partial_t \mathbb{E}_{P_{0|t}(\cdot|y)}(\nabla V)]|_{y=x_t}\|^2 \\
&= \Big\|\lim_{\Delta t \to 0} \frac{1}{\Delta t}\big[\mathbb{E}_{P_{0|t+\Delta t}(\cdot|y)}[\nabla V] - \mathbb{E}_{P_{0|t}(\cdot|y)}[\nabla V]\big]\big|_{y=x_t}\Big\|^2 \\
&\leq L^2 \liminf_{\Delta t \to 0} \frac{1}{(\Delta t)^2} W_1^2\big(P_{0|t+\Delta t}(\cdot \mid x_t), P_{0|t}(\cdot \mid x_t)\big). \tag{10}
\end{aligned}$$

Now $P_{0|t}(\cdot \mid y)$ has density $p_{0|t}(x \mid y) \propto p_0(x)\,e^{-\frac{\|x-y\|^2}{2t}}$ so if $\|\nabla^2 \ln p_0\|_{\mathsf{op}} \leq L$ and $t \leq \frac{1}{2L}$, then $P_{0|t}$ is $\frac{1}{2t}$-strongly log-concave. By Talagrand's transport cost inequality,

$$W_1^2\big(P_{0|t+\Delta t}(\cdot \mid x_t), P_{0|t}(\cdot \mid x_t)\big) \leq 4t\,\mathsf{KL}\big(P_{0|t+\Delta t}(\cdot \mid x_t) \,\|\, P_{0|t}(\cdot \mid x_t)\big).$$

Plugging this back in (10) and using Fatou's lemma and the chain rule for KL,

$$\begin{aligned}
\mathbb{E}[\|[\partial_t \nabla \ln p_t(y)]|_{y=x_t}\|^2] &\leq L^2 \,\mathbb{E} \liminf_{\Delta t \to 0} \frac{1}{(\Delta t)^2}\, 4t\,\mathsf{KL}\big(P_{0|t+\Delta t}(\cdot \mid x_t) \,\|\, P_{0|t}(\cdot \mid x_t)\big) \\
&\leq L^2 \liminf_{\Delta t \to 0} \frac{1}{(\Delta t)^2}\, 4t\,\mathbb{E}\,\mathsf{KL}\big(P_{0|t+\Delta t}(\cdot \mid x_t) \,\|\, P_{0|t}(\cdot \mid x_t)\big) \\
&\leq L^2 \liminf_{\Delta t \to 0} \frac{1}{(\Delta t)^2}\, 4t\,\mathsf{KL}(P_{0,t+\Delta t} \,\|\, P_{0,t}). \tag{11}
\end{aligned}$$

Now

$$\begin{aligned}
\mathsf{KL}(P_{0,t+\Delta t} \,\|\, P_{0,t}) &= \mathbb{E}_{x \sim P_0}\,\mathsf{KL}\big(P_{t+\Delta t|0}(\cdot \mid x), P_{t|0}(\cdot \mid x)\big) = \mathsf{KL}\big(\mathcal{N}(0, (t+\Delta t)I) \,\|\, \mathcal{N}(0, tI)\big) \\
&= \frac{d}{2}\Big(-\ln \frac{t+\Delta t}{t} + \frac{t+\Delta t}{t} - 1\Big) = \frac{d}{4}\big(\frac{\Delta t}{t}\big)^2 + O\big(\big(\frac{\Delta t}{t}\big)^3\big).
\end{aligned}$$

Plugging into (11) gives

$$\mathbb{E}[\|[\partial_t \nabla \ln p_t(y)]|_{y=x_t}\|^2] \leq \frac{L^2 d}{t}. \tag{12}$$

For the second term, by assumption we have $\|\nabla^2 \ln p_t\|_{\mathsf{op}} \leq L$. Then, since $x_t \sim p_t$,

$$\mathbb{E}[\|\nabla^2 \ln p_t(x_t)\,\nabla \ln p_t(x_t)\|^2] \leq L^2\,\mathbb{E}_{p_t}[\|\nabla \ln p_t\|^2] \leq L^3 d \tag{13}$$

using the fact $\mathbb{E}_\mu[\|\nabla \ln \mu\|^2] \leq Ld$ for any measure $\mu$ such that $\ln \mu$ is $L$-smooth, which follows from integration by parts. From (9), (12), and (13), and the elementary inequality $\langle a, b\rangle \leq \|a\|^2 + 4\|b\|^2$, we get

$$\begin{aligned}
\mathbb{E}[\|\partial_t \nabla \ln p_t(x_t)\|^2] &\leq \mathbb{E}[\|[\partial_t \nabla \ln p_t(y)]|_{y=x_t}\|^2] + \mathbb{E}[\|\nabla^2 \ln p_t(x_t)\,\nabla \ln p_t(x_t)\|^2] \\
&\leq L^2 d\big(L + \frac{1}{t}\big). \qquad \square
\end{aligned}$$

The above result holds for the dynamics $\dot{x}_t = -\frac{1}{2}\nabla \ln p_t(x_t)$ for which $(p_t)_{t \geq 0}$ follows the heat flow; this corresponds to the variance-exploding SGM. In this paper, since we wish to consider the SGM based on the variance-conserving Ornstein–Uhlenbeck (OU) process, we can apply the following reparameterization lemma.

**Lemma 2** (Reparameterization). *Suppose that $(x_t)_{t \geq 0}$ satisfies the probability flow ODE for Brownian motion starting at $p_0$; that is, letting $p_t = p_0 * \mathcal{N}(0, tI)$, we have $x_0 \sim p_0$, $\dot{x}_t = -\frac{1}{2} \nabla \ln p_t(x_t)$. Then, if we set*

$$y_t = e^{-t} x_{e^{2t}-1},$$

*then $(y_t)_{t \geq 0}$ satisfies the probability flow ODE for the OU process starting at $p_0$; that is, letting $q_t^{\rightarrow}$ be the density of the OU process at time $t$, we have $y_0 \sim p_0 = q_0^{\rightarrow}$, $\dot{y}_t = -y_t - \nabla \ln q_t^{\rightarrow}(y_t)$.*

*Proof.* By direct calculation, one can check that for any $y \in \mathbb{R}^d$, it holds that $q_t^{\rightarrow}(y) \propto p_{e^{2t}-1}(e^t y)$. The claim follows from the chain rule. $\qquad \square$

**Lemma 3** (Score perturbation for OU). *Suppose $q_t^{\rightarrow}$ is the density of the OU process at time $t$, started at $q_0^{\rightarrow}$, and $y_0 \sim q_0^{\rightarrow}$, $\dot{y}_t = -y_t - \nabla \ln q_t^{\rightarrow}(y_t)$. Suppose for all $t$ and all $x$ that $\|\nabla^2 \ln q_t^{\rightarrow}(x)\|_{\mathsf{op}} \leq L$, where $L \geq 1$. Then,*

$$\mathbb{E}[\|\partial_t \nabla \ln q_t^{\rightarrow}(y_t)\|^2] \lesssim L^2 d \left( L \vee \frac{1}{t} \right).$$

*Proof.* Using the relationship $q_t^{\rightarrow}(y) \propto p_{e^{2t}-1}(e^t y)$,

$$\nabla \ln q_t^{\rightarrow}(y) = e^t \nabla \ln p_{e^{2t}-1}(e^t y),$$

$$\partial_t \nabla \ln q_t^{\rightarrow}(y_t) = \underbrace{e^t \nabla \ln p_{e^{2t}-1}(x_{e^{2t}-1})}_{=:A} + \underbrace{e^t \partial_s \nabla \ln p_s(x_s)|_{s=e^{2t}-1} \cdot 2e^{2t}}_{=:B}.$$

If $\|\nabla^2 \ln q_t^{\rightarrow}\|_{\mathsf{op}} \leq L$, then $\|\nabla^2 \ln p_{e^{2t}-1}\|_{\mathsf{op}} \leq e^{-2t} L$. By Lemma 1,

$$\mathbb{E}[\|\partial_s \nabla \ln p_s(x_s)|_{s=e^{2t}-1}\|^2] \lesssim e^{-4t} L^2 d \left( e^{-2t} L \vee \frac{1}{e^{2t}-1} \right).$$

Hence

$$\mathbb{E}[B^2] \lesssim L^2 d \left( L \vee \frac{1}{t} \right).$$

Next,

$$\mathbb{E}[A^2] \leq e^{2t} \mathbb{E}[\|\nabla \ln p_{e^{2t}-1}(x_{e^{2t}-1})\|^2] \leq e^{2t} e^{-2t} L d \leq L d.$$

The result follows. $\qquad \square$

Finally, we use Lemma 3 to derive a bound on how much the score changes along the trajectory of the probability flow ODE.

**Corollary 1.** *Consider the setting of Lemma 3, and suppose $0 < s < t$, $h = t - s$.*

1. *If $s, t \gtrsim 1/L$, then*

$$\mathbb{E}\left[ \|\nabla \ln q_t^{\rightarrow}(x_t) - \nabla \ln q_s^{\rightarrow}(x_s)\|^2 \right] \lesssim L^3 d h^2.$$

2. *If $\frac{t}{2} \leq s \leq t \lesssim \frac{1}{L}$, then*

$$\mathbb{E}\left[ \|\nabla \ln q_t^{\rightarrow}(x_t) - \nabla \ln q_s^{\rightarrow}(x_s)\|^2 \right] \lesssim \frac{L^2 d h^2}{t}.$$

*Proof.* By Lemma 3,

$$\mathbb{E}\left[ \|\nabla \ln q_t^{\rightarrow}(x_t) - \nabla \ln q_s^{\rightarrow}(x_s)\|^2 \right] = \mathbb{E}\left[ \left\| \int_s^t \partial_u \nabla \ln q_u^{\rightarrow}(x_u) \, \mathrm{d}u \right\|^2 \right]$$

$$\leq (t - s) \int_s^t \mathbb{E}[\|\partial_u \nabla \ln q_u^{\rightarrow}(x_u)\|^2] \, \mathrm{d}u$$

$$\lesssim h \int_s^t L^2 d \max\left\{ L, \frac{1}{u} \right\} \, \mathrm{d}u.$$

In the first case, this is bounded by $O(L^3 d h^2)$. In the second case, this is bounded by $O(L^2 d h \int_s^t \frac{1}{u} \, \mathrm{d}u) = O(L^2 d h \ln(t/s)) = O(L^2 d h^2 / t)$. $\qquad \square$

# C Predictor step

Next, we need an ODE discretization analysis.

**Lemma 4.** *Suppose the score function satisfies Assumption 2. Assume that $L \geq 1$, $h \lesssim 1/L$, and $T - (t_0 + h) \geq \frac{T-t_0}{2}$. Then*

$$W_2(qP_{\mathsf{ODE}}^{t_0,h}, q\widehat{P}_{\mathsf{ODE}}^{t_0,h}) \lesssim Ld^{1/2}h^2 \left(L^{1/2} \vee \frac{1}{(T-t_0)^{1/2}}\right) + h\varepsilon_{\mathsf{sc}}.$$

*Proof.* We have the ODEs

$$\dot{x}_t = x_t + \nabla \ln q_t(x_t),$$
$$\dot{\widehat{x}}_t = \widehat{x}_t + s_{t_0}(\widehat{x}_{t_0}),$$

for $t_0 \leq t \leq t_0 + h$, with $x_{t_0} = \widehat{x}_{t_0} \sim q$, $x_{t_0+h} \sim qP_{\mathsf{ODE}}$, and $\widehat{x}_{t_0+h} \sim q\widehat{P}_{\mathsf{ODE}}$. Then,

$$
\begin{aligned}
\partial_t \|x_t - \widehat{x}_t\|^2 &= 2 \langle x_t - \widehat{x}_t, \dot{x}_t - \dot{\widehat{x}}_t \rangle \\
&= 2 \left( \|x_t - \widehat{x}_t\|^2 + \langle x_t - \widehat{x}_t, \nabla \ln q_t(x_t) + s_{t_0}(\widehat{x}_{t_0}) \rangle \right) \\
&\leq \left(2 + \frac{1}{h}\right) \|x_t - \widehat{x}_t\|^2 + h \|\nabla \ln q_t(x_t) - s_{t_0}(\widehat{x}_{t_0})\|^2.
\end{aligned}
$$

By Grönwall's inequality, noting that $h = O(1)$,

$$
\begin{aligned}
\mathbb{E}[\|x_{t_0+h} - \widehat{x}_{t_0+h}\|^2] &\leq \exp\left(\left(2 + \frac{1}{h}\right)h\right) \int_{t_0}^{t_0+h} h \, \mathbb{E}[\|\nabla \ln q_t(x_t) - s_{t_0}(\widehat{x}_{t_0})\|^2] \, dt \\
&\lesssim h \int_{t_0}^{t_0+h} \mathbb{E}[\|\nabla \ln q_t(x_t) - s_{t_0}(\widehat{x}_{t_0})\|^2] \, dt.
\end{aligned}
\tag{14}
$$

We split up the error term as

$$\|\nabla \ln q_t(x_t) - s_{t_0}(\widehat{x}_{t_0})\|^2 \lesssim \|\nabla \ln q_t(x_t) - \nabla \ln q_{t_0}(x_{t_0})\|^2 + \|\nabla \ln q_{t_0}(x_{t_0}) - s_{t_0}(\widehat{x}_{t_0})\|^2.$$

By Corollary 1, the expectation of the first term is bounded by

$$\mathbb{E}[\|\nabla \ln q_t(x_t) - \nabla \ln q_{t_0}(x_{t_0})\|^2] \lesssim L^2 dh^2 \left(L \vee \frac{1}{T-t_0}\right).$$

The second term is bounded in expectation by $\varepsilon_{\mathsf{sc}}^2$. Plugging back into (14) gives

$$\mathbb{E}[\|x_{t_0+h} - \widehat{x}_{t_0+h}\|^2] \lesssim h^2 \left(L^2 dh^2 \left(L \vee \frac{1}{T-t_0}\right) + \varepsilon_{\mathsf{sc}}^2\right).$$

The Wasserstein distance is bounded by the square root of this quantity, and the lemma follows. $\square$

Lemma 4 suggests that focusing on the dependence on $d$, we will be able to take $h \asymp d^{-1/2}$ (we need to keep one factor of $h$ in the bound, as we need to sum up the bound over $1/h$ iterations).

*Remark* 2. Our improved score perturbation lemma is necessary to obtain this $d^{1/2}$ dependence. The original score perturbation lemma [LLT22, Lemma C.11–12] combined with a space discretization bound gives a bound of

$$\mathbb{E}[\|\nabla \ln q_t^{\rightarrow}(x_t) - \nabla \ln q_s^{\rightarrow}(x_s)\|^2] \lesssim L^2 dh$$

in place of Corollary 1. Note this is a $\frac{1}{2}$-Hölder continuity bound rather than a Lipschitz bound. The bound in Lemma 4 then becomes

$$W_2(qP_{\mathsf{ODE}}^{t_0,h}, q\widehat{P}_{\mathsf{ODE}}^{t_0,h}) \lesssim Ld^{1/2}h^{3/2} + h\varepsilon_{\mathsf{sc}},$$

and we would only be able to take $h \asymp d^{-1}$. We also note that our bound has an extra factor of $\max\{L^{1/2}, (T-t_0)^{-1/2}\}$; we do not know if this extra factor is necessary.

We now iterate Lemma 4 to obtain the following result. Note that we now need to also assume that the score estimate is $L$-Lipschitz.

**Lemma 5.** *Suppose that both Assumptions 2 and 3 hold. Let $h_1, \ldots, h_N > 0$ be a sequence such that letting $t_N = h_1 + \cdots + h_N$, we have $t_N \leq 1/L$. Let $h_{\max} = \max_{1 \leq n \leq N} h_n$.*

1. *If $T - (t_0 + t_N) \gtrsim 1/L$, then*

$$W_2(qP_{\mathsf{ODE}}^{t_0,h_1,\ldots,h_N}, q\widehat{P}_{\mathsf{ODE}}^{t_0,h_1,\ldots,h_N}) \lesssim L^{3/2}d^{1/2}h_{\max}t_N + \varepsilon_{\mathsf{sc}}t_N \leq L^{1/2}d^{1/2}h_{\max} + \frac{\varepsilon_{\mathsf{sc}}}{L} .$$

2. *If $T - t_0 \lesssim 1/L$ and $h_{n+1} \leq \frac{T-t_0-t_n}{2}$ for each $n$, then*

$$W_2(qP_{\mathsf{ODE}}^{t_0,h_1,\ldots,h_N}, q\widehat{P}_{\mathsf{ODE}}^{t_0,h_1,\ldots,h_N}) \lesssim L^{1/2}d^{1/2}h_{\max} + \varepsilon_{\mathsf{sc}}t_N \leq L^{1/2}d^{1/2}h_{\max} + \frac{\varepsilon_{\mathsf{sc}}}{L} .$$

*Proof.* We abbreviate $P_{\mathsf{ODE}}^N := P_{\mathsf{ODE}}^{t_0,h_1,\ldots,h_N}$ and $\widehat{P}_{\mathsf{ODE}}^N := \widehat{P}_{\mathsf{ODE}}^{t_0,h_1,\ldots,h_N}$. Using the triangle inequality,

$$W_2(qP_{\mathsf{ODE}}^N, q\widehat{P}_{\mathsf{ODE}}^N) \leq W_2(qP_{\mathsf{ODE}}^N, qP_{\mathsf{ODE}}^{N-1}\widehat{P}_{\mathsf{ODE}}) + W_2(qP_{\mathsf{ODE}}^{N-1}\widehat{P}_{\mathsf{ODE}}, q\widehat{P}_{\mathsf{ODE}}^N)$$
$$\leq O(Ld^{1/2}h_N^2 \max\{L^{1/2}, (T - t_0 - t_N)^{-1/2}\} + h_N\varepsilon_{\mathsf{sc}})$$
$$+ \exp(O(Lh_N))\, W_2(qP_{\mathsf{ODE}}^{N-1}, q\widehat{P}_{\mathsf{ODE}}^{N-1})$$

where the bound on the first term is by Lemma 4. By induction,

$$W_2(qP_{\mathsf{ODE}}^N, q\widehat{P}_{\mathsf{ODE}}^N) \lesssim \sum_{n=1}^N \left(Ld^{1/2}h_n^2 \max\{L^{1/2}, (T - t_0 - t_n)^{-1/2}\} + h_n\varepsilon_{\mathsf{sc}}\right)$$
$$\times \exp(O(L\,(h_{n+1} + \cdots + h_N))) .$$

By assumption, $h_{n+1} + \cdots + h_N \leq t_N \leq 1/L$. In the first case, we get

$$W_2(qP_{\mathsf{ODE}}^N, q\widehat{P}_{\mathsf{ODE}}^N) \lesssim L^{3/2}d^{1/2}h_{\max}t_N + \varepsilon_{\mathsf{sc}}t_N .$$

In the second case we get

$$W_2(qP_{\mathsf{ODE}}^N, q\widehat{P}_{\mathsf{ODE}}^N) \lesssim Ld^{1/2}h_{\max} \sum_{n=1}^N \frac{h_n}{(T - t_0 - t_n)^{1/2}} + \varepsilon_{\mathsf{sc}}t_N \lesssim L^{1/2}d^{1/2}h_{\max} + \varepsilon_{\mathsf{sc}}t_N$$

by interpreting the summation as a Riemann sum, and noting that the condition $h_{n+1} \leq \frac{T-t_0-t_n}{2}$ implies that this is a constant-factor approximation of the integral $\int_{T-t_0-t_N}^{T-t_0} \frac{1}{t^{1/2}}\, dt \lesssim \sqrt{T - t_0}$. □

**Choice of step sizes.** In the first case, we can take all the step sizes to be equal, but in the second case, we may need to take decreasing step sizes. Given a target time $T - t_0 - t_N = \delta$, by taking $h_1 = h_{\max}$ and then

$$h_n = \min\left\{\delta, h_{\max}, \frac{T - t_0 - t_{n-1}}{2}\right\},$$

we can reach the target time in

$$N = O\left(\frac{1}{Lh_{\max}} + \ln \frac{h_{\max}}{\delta}\right) \quad \text{steps} .$$

# D  Corrector step

In §D.1 (resp. §D.2), we will show that if $p$, $q$ are close in Wasserstein distance, then running the corrector step based on the overdamped (resp. underdamped) Langevin diffusion starting from $p$ and from $q$ for some amount of time results in distributions which are close in *total variation* distance. In the end-to-end analysis in §E, we combine this "total variation to Wasserstein" regularization with the Wasserstein discretization analysis of the predictor step in §C in order to establish our final results.

### D.1 Corrector via overdamped Langevin

We will take the potential and score estimate defining the Markov kernels $P_{\mathsf{LD}}$ and $\widehat{P}_{\mathsf{LMC}}$ from §2.3 to be $U$ and $s$ respectively. Recall that these correspond respectively to running the overdamped Langevin diffusion with stationary distribution $q \propto \exp(-U)$ and running the discretized diffusion with score estimate $s$, both for time $h$.

The main result of this section is to show that $p\widehat{P}_{\mathsf{LMC}}^N$ and $q$ are close in total variation if $p$ and $q$ are close in Wasserstein.

**Theorem 4** (Overdamped corrector). *For any* $T_{\mathsf{corr}} \coloneqq Nh \lesssim 1/L$,

$$\mathsf{TV}(p\widehat{P}_{\mathsf{LMC}}^N, q) \lesssim W_2(p, q)/\sqrt{T_{\mathsf{corr}}} + \varepsilon_{\mathsf{sc}}\sqrt{T_{\mathsf{corr}}} + L\sqrt{dhT_{\mathsf{corr}}} \,.$$

*In particular, for* $T_{\mathsf{corr}} \asymp 1/L$,

$$\mathsf{TV}(p\widehat{P}_{\mathsf{LMC}}^N, q) \lesssim \sqrt{L}\,W_2(p, q) + \varepsilon_{\mathsf{sc}}/\sqrt{L} + \sqrt{Ldh} \,.$$

We will bound $\mathsf{TV}(pP_{\mathsf{LD}}^N, q)$ and $\mathsf{TV}(p\widehat{P}_{\mathsf{LMC}}^N, pP_{\mathsf{LD}}^N)$ separately. For the former, we use the following short-time regularization result:

**Lemma 6** ([BGL01, Lemma 4.2]). *If* $T_{\mathsf{corr}} \lesssim 1/L$, *then*

$$\mathsf{TV}(pP_{\mathsf{LD}}^N, q) \lesssim \sqrt{\mathsf{KL}(pP_{\mathsf{LD}}^N \parallel q)} \lesssim W_2(p, q)/\sqrt{T_{\mathsf{corr}}} \,.$$

*Proof.* The first inequality is Pinsker's inequality. The second inequality is a consequence of [BGL01, Lemma 4.2], which gives a bound of $\mathsf{KL}(pP_{\mathsf{LD}}^N \parallel q) \lesssim L\,(1 + 1/(e^{2LT_{\mathsf{corr}}} - 1))\,W_2^2(p, q)$. The claim then follows from simplifying by using $T_{\mathsf{corr}} \lesssim 1/L$. $\qquad\square$

For the latter term, we introduce notation for two stochastic processes.

$$\begin{aligned}
\mathrm{d}x_t^\circ &= -\nabla U(x_t^\circ)\,\mathrm{d}t + \sqrt{2}\,\mathrm{d}B_t\,, & x_0^\circ &\sim q\,, \\
\mathrm{d}x_t &= -\nabla U(x_t)\,\mathrm{d}t + \sqrt{2}\,\mathrm{d}B_t\,, & x_0 &\sim p\,.
\end{aligned}$$

Note that for any integer $k \geq 0$,

$$x_{kh}^\circ \sim qP_{\mathsf{LD}}^k\,, \qquad x_{kh} \sim pP_{\mathsf{LD}}^k\,.$$

Observe that marginally, $qP_{\mathsf{LD}}^k = q$ for any $k \geq 0$ because $q$ is the stationary distribution of the Langevin diffusion. The three processes are coupled by using the same Brownian motion and by coupling $x_0 = \widehat{x}_0 \sim p$ and $x_0^\circ \sim q$ optimally.

Before we proceed to bound $\mathsf{TV}(pP_{\mathsf{LD}}^N, p\widehat{P}_{\mathsf{LMC}}^N)$, we need the following simple lemma.

**Lemma 7.** *If* $T_{\mathsf{corr}} \lesssim 1/L$, *then*

$$\mathbb{E}[\|x_t - x_t^\circ\|^2] \lesssim W_2^2(p, q)$$

*for all* $0 \leq t \leq T_{\mathsf{corr}}$.

*Proof.* By Itô's formula,

$$\mathrm{d}(\|x_t - x_t^\circ\|^2) = -2\,\langle x_t - x_t^\circ, \nabla U(x_t) - \nabla U(x_t^\circ)\rangle \leq 2L\,\|x_t - x_t^\circ\|^2\,.$$

By Grönwall's inequality,

$$\|x_t - x_t^\circ\|^2 \leq e^{2Lt}\,\|x_0 - x_0^\circ\|^2\,,$$

so that if we couple the two processes by coupling $x_0$ and $x_0^\circ$ optimally, we conclude that

$$\mathbb{E}[\|x_t - x_t^\circ\|^2] \leq e^{2Lt}\,\mathbb{E}[\|x_0 - x_0^\circ\|^2] = e^{2Lt}\,W_2^2(p, q) \lesssim W_2^2(p, q)\,,$$

recalling that $t \leq T_{\mathsf{corr}} \lesssim 1/L$ by hypothesis. $\qquad\square$

It remains to bound $\mathsf{TV}(p\widehat{P}_{\mathsf{LMC}}^N, pP_{\mathsf{LD}}^N)$.

**Lemma 8.** *If $T_{\mathsf{corr}} \lesssim 1/L$, then*

$$\mathsf{TV}(p\widehat{P}^N_{\mathsf{LMC}}, pP^N_{\mathsf{LD}}) \lesssim \sqrt{\mathsf{KL}(pP^N_{\mathsf{LD}} \,\|\, p\widehat{P}^N_{\mathsf{LMC}})} \lesssim L\sqrt{T_{\mathsf{corr}}}\, W_2(p,q) + \varepsilon_{\mathsf{sc}}\sqrt{T_{\mathsf{corr}}} + L\sqrt{dhT_{\mathsf{corr}}}\,.$$

*Proof.* As $x$ and $\widehat{x}$ are driven by the same Brownian motion, by Girsanov's theorem[8] and the data processing inequality we have

$$\mathsf{KL}(pP^N_{\mathsf{LD}} \,\|\, p\widehat{P}^N_{\mathsf{LMC}}) \lesssim \sum_{k=0}^{N-1} \int_{kh}^{(k+1)h} \mathbb{E}[\|s(x_{kh}) - \nabla U(x_u)\|^2]\,\mathrm{d}u\,.$$

We can decompose the integrand as follows:

$$\mathbb{E}[\|s(x_{kh}) - \nabla U(x_u)\|^2] \lesssim \mathbb{E}\big[\|s(x_{kh}) - s(x_{kh}^\circ)\|^2 + \|s(x_{kh}^\circ) - \nabla U(x_{kh}^\circ)\|^2$$
$$+ \|\nabla U(x_{kh}^\circ) - \nabla U(x_u^\circ)\|^2 + \|\nabla U(x_u^\circ) - \nabla U(x_u)\|^2\big]$$
$$\leq L^2\,\mathbb{E}[\|x_{kh} - x_{kh}^\circ\|^2] + \varepsilon_{\mathsf{sc}}^2 + L^2\,\mathbb{E}[\|x_{kh}^\circ - x_u^\circ\|^2] + L^2\,\mathbb{E}[\|x_u^\circ - x_u\|^2]$$

$$\lesssim L^2\, W_2^2(p,q) + \varepsilon_{\mathsf{sc}}^2 + L^2\,\mathbb{E}[\|x_{kh}^\circ - x_u^\circ\|^2] \tag{15}$$

where we used Lemma 7 to bound $\mathbb{E}[\|x_u^\circ - x_u\|^2]$.

It remains to bound $\mathbb{E}[\|x_{kh}^\circ - x_u^\circ\|^2]$. Note that

$$\mathbb{E}[\|x_{kh}^\circ - x_u^\circ\|^2] = \mathbb{E}\Big[\Big\|\int_{kh}^u -\nabla U(x_s^\circ)\,\mathrm{d}s + \sqrt{2}\,(B_u - B_{kh})\Big\|^2\Big] \lesssim h\int_{kh}^u \mathbb{E}[\|\nabla U(x_s^\circ)\|^2]\,\mathrm{d}s + dh$$
$$\leq Ldh^2 + dh \lesssim dh\,,$$

where in the last step we used that $\mathbb{E}[\|\nabla U(x_u^\circ)\|^2] \leq Ld$. Substituting this into (15), we obtain

$$\mathsf{KL}(pP^N_{\mathsf{LD}} \,\|\, p\widehat{P}^N_{\mathsf{LMC}}) \lesssim L^2 T_{\mathsf{corr}}\, W_2^2(p,q) + \varepsilon_{\mathsf{sc}}^2 T_{\mathsf{corr}} + L^2 dh T_{\mathsf{corr}}\,.$$

The claimed bound on $\mathsf{TV}(pP^N_{\mathsf{LD}}, p\widehat{P}^N_{\mathsf{LMC}})$ follows by Pinsker's inequality. $\qquad\square$

*Proof of Theorem 4.* This is immediate from Lemma 6 and Lemma 8, recalling that $T_{\mathsf{corr}} \lesssim 1/L$ so that the bound in Lemma 6 dominates the $W_2(p,q)$ term in Lemma 8. $\qquad\square$

### D.2  Corrector via underdamped Langevin

Throughout, we set the friction parameter to

$$\gamma \asymp \sqrt{L}\,.$$

We will take the potential and score estimate defining the Markov kernels $P_{\mathsf{ULD}}$ and $\widehat{P}_{\mathsf{ULMC}}$ from §2.3 to be $U$ and $s$ respectively. Recall that these correspond respectively to running the underdamped Langevin diffusion with stationary distribution $q$ and running the discretized diffusion with score estimate $s$, both for time $h$.

Given probability measures $p$ and $q$, we write $\boldsymbol{p} := p \otimes \gamma_d$ and $\boldsymbol{q} := q \otimes \gamma_d$, where $\gamma_d$ is the standard Gaussian measure in $\mathbb{R}^d$.

The main result of this section is to show that $\boldsymbol{p}\widehat{P}^N_{\mathsf{ULMC}}$ and $\boldsymbol{q}$ are close in total variation if $p$ and $q$ are close in Wasserstein. Compared to §D.1, the discretization error for the underdamped Langevin diffusion is smaller.

**Theorem 5** (Underdamped corrector). *For $T_{\mathsf{corr}} \lesssim 1/\sqrt{L}$,*

$$\mathsf{TV}(\boldsymbol{p}\widehat{P}^N_{\mathsf{ULMC}}, \boldsymbol{q}) \lesssim \frac{W_2(p,q)}{L^{1/4} T_{\mathsf{corr}}^{3/2}} + \frac{\varepsilon_{\mathsf{sc}} T_{\mathsf{corr}}^{1/2}}{L^{1/4}} + L^{3/4} T_{\mathsf{corr}}^{1/2} d^{1/2} h\,.$$

*In particular, if we take $T_{\mathsf{corr}} \asymp 1/\sqrt{L}$, then*

$$\mathsf{TV}(\boldsymbol{p}\widehat{P}^N_{\mathsf{ULMC}}, \boldsymbol{q}) \lesssim \sqrt{L}\, W_2(p,q) + \varepsilon_{\mathsf{sc}}/\sqrt{L} + \sqrt{L d}\, h\,.$$

---

[8] Although the validity of Girsanov's theorem typically requires Novikov's condition to be satisfied, this can be avoided via an approximation argument as in [Che+23a].

We will bound $\mathsf{TV}(\boldsymbol{p}P_{\mathsf{ULD}}^N, \boldsymbol{q})$ and $\mathsf{TV}(\boldsymbol{p}\widehat{P}_{\mathsf{ULMC}}^N, \boldsymbol{p}P_{\mathsf{ULD}}^N)$ separately. For the former, we use the short-time regularization result of [GW12]:

**Lemma 9.** *If* $T_{\mathsf{corr}} \lesssim 1/\sqrt{L}$*, then*

$$\mathsf{TV}(\boldsymbol{p}P_{\mathsf{ULD}}^N, \boldsymbol{q}) \lesssim \sqrt{\mathsf{KL}(\boldsymbol{p}P_{\mathsf{ULD}}^N \,\|\, \boldsymbol{q})} \lesssim \frac{W_2(p, q)}{L^{1/4}T_{\mathsf{corr}}^{3/2}} \,.$$

*Proof.* This is a consequence of [GW12, Corollary 4.7 (1)]. The condition to check therein is their Eq. (3.6), which in our setting is satisfied by the constants $K_1 = L$ and $K_2 = \gamma$. The Corollary then states that for the cost function

$$c_{T_{\mathsf{corr}}}((z, v), (z', v')) := \inf_{t \in (0, T_{\mathsf{corr}}]} \frac{t}{2\gamma} \left\{ \left( \frac{6}{t^2} + L + \frac{3\gamma}{2t} \right) \|z - z'\| + \left( \frac{4}{t} + \frac{4Lt}{27} + \gamma \right) \|v - v'\| \right\}^2 \,,$$

we have $\mathsf{KL}(\boldsymbol{p}P_{\mathsf{ULD}}^N \,\|\, \boldsymbol{q}) \leq W_{c_{T_{\mathsf{corr}}}}(p \otimes \gamma_d, q \otimes \gamma_d)$. For $v = v'$ and $T_{\mathsf{corr}} \lesssim 1/\sqrt{L}$, note that

$$c_{T_{\mathsf{corr}}}((z, v), (z', v)) \lesssim \frac{1}{L^{1/2}T_{\mathsf{corr}}^3} \|z - z'\|^2 \,,$$

so the claim follows by Pinsker's inequality. $\qquad\square$

Next, we define the following processes: $dz_t^\circ = v_t^\circ \, dt$, $dz_t = v_t \, dt$,

$$\begin{aligned}
dv_t^\circ &= -\gamma v_t^\circ \, dt - \nabla U(z_t^\circ) \, dt + \sqrt{2\gamma} \, dB_t \,, & (z_0^\circ, v_0^\circ) &\sim \boldsymbol{q} \,, \\
dv_t &= -\gamma v_t \, dt - \nabla U(z_t) \, dt + \sqrt{2\gamma} \, dB_t \,, & (z_0, v_0) &\sim \boldsymbol{p} \,.
\end{aligned}$$

It follows that for any integer $k \geq 0$,

$$(z_{kh}^\circ, v_{kh}^\circ) \sim \boldsymbol{q}P_{\mathsf{ULD}}^k = \boldsymbol{q} \,, \qquad (z_{kh}, v_{kh}) \sim \boldsymbol{p}P_{\mathsf{ULD}}^k \,.$$

We couple these processes by using the same Brownian motion and coupling $q \otimes \gamma_d$ and $p \otimes \gamma_d$ optimally (in particular, $v_0 = v_0^\circ$).

Before we proceed to bound $\mathsf{TV}(pP_{\mathsf{ULD}}^N, p\widehat{P}_{\mathsf{ULMC}}^N)$, we start with the following lemma.

**Lemma 10.** *If* $T_{\mathsf{corr}} \lesssim 1/\sqrt{L}$*, then for all* $0 \leq t \leq T_{\mathsf{corr}}$*,*

$$\mathbb{E}[\|z_t - z_t^\circ\|^2] \lesssim W_2^2(p, q) \,.$$

*Proof.* We have

$$\nabla U(z_t) - \nabla U(z_t^\circ) = \left( \int_0^1 \nabla^2 U(z_t + u \, (z_t^\circ - z_t)) \, du \right) (z_t - z_t^\circ) := \mathcal{H}_t(z_t - z_t^\circ) \,,$$

and the operator $\mathcal{H}_t$ satisfies

$$\|\mathcal{H}_t\|_{\mathsf{op}} \leq L \,.$$

Let $\alpha := 2/\gamma$. For the vectors $\delta_t := (z_t + \alpha v_t) - (z_t^\circ + \alpha v_t^\circ)$ and $\eta_t := z_t - z_t^\circ$, we have

$$\begin{aligned}
\frac{1}{2} \, d(\|\delta_t\|^2 + \|\eta_t\|^2) &= -(\delta_t, \eta_t)^\intercal \begin{bmatrix} (\gamma - \frac{1}{\alpha}) I_d & \frac{1}{2} (\alpha \mathcal{H}_t - \gamma \, I_d) \\ \frac{1}{2} (\alpha \mathcal{H}_t - \gamma \, I_d) & \frac{1}{\alpha} I_d \end{bmatrix} (\delta_t, \eta_t) \\
&\lesssim \sqrt{L} \, (\|\delta_t\|^2 + \|\eta_t\|^2) \,.
\end{aligned}$$

By Grönwall's inequality,

$$\|\delta_t\|^2 + \|\eta_t\|^2 \leq e^{O(\sqrt{L}t)} \, (\|\delta_0\|^2 + \|\eta_0\|^2) \,,$$

so if we couple the two processes by coupling $z_0$ and $z_0^\circ$ optimally and taking $v_0 = v_0^\circ$, we obtain

$$\mathbb{E}[\|z_t - z_t^\circ\|^2] \lesssim \mathbb{E}[\|\delta_t\|^2 + \|\eta_t\|^2] \leq e^{O(\sqrt{L}t)} \, \mathbb{E}[\|\delta_0\|^2 + \|\eta_0\|^2] \lesssim e^{O(\sqrt{L}t)} \, W_2^2(p, q) \lesssim W_2^2(p, q) \,,$$

recalling that $t \leq T_{\mathsf{corr}} \lesssim 1/\sqrt{L}$ by hypothesis. $\qquad\square$

It remains to bound $\mathsf{TV}(\boldsymbol{p}\widehat{P}_{\mathsf{ULMC}}^N, \boldsymbol{p}P_{\mathsf{ULD}}^N)$.

**Lemma 11.** *If $T_{\mathsf{corr}} \lesssim 1/\sqrt{L}$, then*

$$\mathsf{TV}(\boldsymbol{p}\widehat{P}^N_{\mathsf{ULMC}}, \boldsymbol{p}P^N_{\mathsf{ULD}}) \lesssim \sqrt{\mathsf{KL}(\boldsymbol{p}P^N_{\mathsf{ULD}} \,\|\, \boldsymbol{p}\widehat{P}^N_{\mathsf{ULMC}})}$$
$$\lesssim L^{3/4}T^{1/2}_{\mathsf{corr}} W_2(p,q) + L^{-1/4}T^{1/2}_{\mathsf{corr}}\varepsilon_{\mathsf{sc}} + L^{3/4}T^{1/2}_{\mathsf{corr}}d^{1/2}h\,.$$

*Proof.* As $(z,v)$ and $(z^\circ, v^\circ)$ are driven by the same Brownian motion, by Girsanov's theorem[9] and the data processing inequality we have

$$\mathsf{KL}(\boldsymbol{p}P^N_{\mathsf{ULD}} \,\|\, \boldsymbol{p}\widehat{P}^N_{\mathsf{ULMC}}) \lesssim \frac{1}{\gamma}\sum_{k=0}^{N-1}\int_{kh}^{(k+1)h} \mathbb{E}[\|s(z_{kh}) - \nabla U(z_u)\|^2]\,\mathrm{d}u\,.$$

We can decompose the integrand as follows:

$$\mathbb{E}[\|s(z_{kh}) - \nabla U(z_u)\|^2] \lesssim \mathbb{E}\big[\|s(z_{kh}) - s(z^\circ_{kh})\|^2 + \|s(z^\circ_{kh}) - \nabla U(z^\circ_{kh})\|^2$$
$$+ \|\nabla U(z^\circ_{kh}) - \nabla U(z^\circ_u)\|^2 + \|\nabla U(z^\circ_u) - \nabla U(z_u)\|^2\big]$$
$$\leq L^2\,\mathbb{E}[\|z_{kh} - z^\circ_{kh}\|^2] + \varepsilon^2_{\mathsf{sc}} + L^2\,\mathbb{E}[\|z^\circ_{kh} - z^\circ_u\|^2] + L^2\,\mathbb{E}[\|z^\circ_u - z_u\|^2]$$

$$\lesssim L^2\,W^2_2(p,q) + \varepsilon^2_{\mathsf{sc}} + L^2\,\mathbb{E}[\|z^\circ_{kh} - z^\circ_u\|^2]\,, \tag{16}$$

where we applied Lemma 10.

It remains to bound $\mathbb{E}[\|z^\circ_{kh} - z^\circ_u\|^2]$. Note that

$$\mathbb{E}[\|z^\circ_{kh} - z^\circ_u\|^2] = \mathbb{E}\Big[\Big\|\int_{kh}^u v^\circ_s\,\mathrm{d}s\Big\|^2\Big] \leq h\int_{kh}^u \mathbb{E}[\|v^\circ_s\|^2]\,\mathrm{d}s \leq dh^2\,,$$

where in the last step we used the fact that $v^\circ_s \sim \gamma_d$. Substituting this into (16), we conclude that

$$\mathsf{KL}(\boldsymbol{p}P^N_{\mathsf{ULD}} \,\|\, \boldsymbol{p}\widehat{P}^N_{\mathsf{ULMC}}) \lesssim L^{3/2}T_{\mathsf{corr}} W^2_2(p,q) + L^{-1/2}T_{\mathsf{corr}}\varepsilon^2_{\mathsf{sc}} + L^{3/2}dh^2 T_{\mathsf{corr}}\,.$$

The claimed bound on $\mathsf{TV}(\boldsymbol{p}P^N_{\mathsf{ULD}}, \boldsymbol{p}\widehat{P}^N_{\mathsf{ULMC}})$ follows by Pinsker's inequality. $\qquad\square$

*Proof of Theorem 5.* This is immediate from Lemma 9 and Lemma 11, recalling that $T_{\mathsf{corr}} \lesssim 1/\sqrt{L}$ so that the bound in Lemma 9 dominates the $W_2(p,q)$ term in Lemma 11. $\qquad\square$

*Remark* 3. In all other sections of this paper, we abuse notation as follows. Given a distribution $p$ on $\mathbb{R}^d$, we write $p\widehat{P}_{\mathsf{ULMC}}$ to denote the projection onto the $z$-coordinate of $\boldsymbol{p}\widehat{P}_{\mathsf{ULMC}}$, i.e., we view $\widehat{P}_{\mathsf{ULMC}}$ as a Markov kernel on $\mathbb{R}^d$ rather than on $\mathbb{R}^d \times \mathbb{R}^d$ (and similarly for $P_{\mathsf{ULD}}$).

## E   End-to-end analysis

**Lemma 12** (TV error after one round of predictor and corrector). *Choose predictor step sizes $h_1, \ldots, h_{N_{\mathsf{pred}}}$ as in Lemma 5 with $T_{\mathsf{pred}} = h_1 + \cdots + h_{N_{\mathsf{pred}}} \leq 1/L$. That is, if $T - t_0 - T_{\mathsf{pred}} \lesssim 1/L$, then we ensure that $h_{n+1} \leq \frac{T-t_0-h_1-\cdots-h_n}{2}$ for all $n$, and if $T - t_0 \gtrsim 1/L$, then we can take $h_1 = \cdots = h_N$. Let $h_{\mathsf{pred}} := \max_{1 \leq n \leq N_{\mathsf{pred}}} h_n$ and abbreviate $P^{(N_{\mathsf{pred}})}_{\mathsf{ODE}} := P^{t_0,h_1,\ldots,h_{N_{\mathsf{pred}}}}_{\mathsf{ODE}}$ (and similarly for $\widehat{P}_{\mathsf{ODE}}$).*

1. *Consider running the overdamped Langevin corrector for time $T_{\mathsf{corr}} \asymp 1/L$, step size $h_{\mathsf{corr}}$, and stationary distribution $q_{t_0}P^{(N_{\mathsf{pred}})}_{\mathsf{ODE}} = q_{t_0+T_{\mathsf{pred}}}$; set $N_{\mathsf{corr}} = T_{\mathsf{corr}}/h_{\mathsf{corr}}$. Then,*

   $$\mathsf{TV}(p\widehat{P}^{(N_{\mathsf{pred}})}_{\mathsf{ODE}}\widehat{P}^{N_{\mathsf{corr}}}_{\mathsf{LMC}}, q_{t_0+T_{\mathsf{pred}}}) \leq \mathsf{TV}(p, q_{t_0}) + O\Big(L\sqrt{d}\,h_{\mathsf{pred}} + \sqrt{Ldh_{\mathsf{corr}}} + \frac{\varepsilon_{\mathsf{sc}}}{\sqrt{L}}\Big)\,.$$

2. *Consider running the underdamped Langevin corrector for time $T_{\mathsf{corr}} \asymp 1/\sqrt{L}$, step size $h_{\mathsf{corr}}$, and stationary distribution $q_{t_0}P^{(N_{\mathsf{pred}})}_{\mathsf{ODE}} = q_{t_0+T_{\mathsf{pred}}}$; set $N_{\mathsf{corr}} = T_{\mathsf{corr}}/h_{\mathsf{corr}}$. Then,*

   $$\mathsf{TV}(p\widehat{P}^{(N_{\mathsf{pred}})}_{\mathsf{ODE}}\widehat{P}^{N_{\mathsf{corr}}}_{\mathsf{ULMC}}, q_{t_0+T_{\mathsf{pred}}}) \leq \mathsf{TV}(p, q_{t_0}) + O\Big(L\sqrt{d}\,h_{\mathsf{pred}} + \sqrt{Ld}\,h_{\mathsf{corr}} + \frac{\varepsilon_{\mathsf{sc}}}{\sqrt{L}}\Big)\,.$$

---

[9]Again, we can avoid checking Novikov's condition using the approximation argument of [Che+23a].

*Proof.* By the triangle inequality and the data-processing inequality,

$$\mathsf{TV}(p\widehat{P}_{\mathsf{ODE}}^{(N_{\mathsf{pred}})}\widehat{P}_{\mathsf{LMC}}^{N_{\mathsf{corr}}}, q_{t_0+T_{\mathsf{pred}}})$$

$$\leq \mathsf{TV}(p\widehat{P}_{\mathsf{ODE}}^{(N_{\mathsf{pred}})}\widehat{P}_{\mathsf{LMC}}^{N_{\mathsf{corr}}}, q_{t_0}\widehat{P}_{\mathsf{ODE}}^{(N_{\mathsf{pred}})}\widehat{P}_{\mathsf{LMC}}^{N_{\mathsf{corr}}}) + \mathsf{TV}(q_{t_0}\widehat{P}_{\mathsf{ODE}}^{(N_{\mathsf{pred}})}\widehat{P}_{\mathsf{LMC}}^{N_{\mathsf{corr}}}, q_{t_0+T_{\mathsf{pred}}})$$

$$\leq \mathsf{TV}(p, q_{t_0}) + \mathsf{TV}(q_{t_0}\widehat{P}_{\mathsf{ODE}}^{(N_{\mathsf{pred}})}\widehat{P}_{\mathsf{LMC}}^{N_{\mathsf{corr}}}, q_{t_0+T_{\mathsf{pred}}}).$$

For overdamped Langevin, applying Theorem 4,

$$\mathsf{TV}(q_{t_0}\widehat{P}_{\mathsf{ODE}}^{(N_{\mathsf{pred}})}\widehat{P}_{\mathsf{LMC}}^{N_{\mathsf{corr}}}, q_{t_0+T_{\mathsf{pred}}}) \lesssim \sqrt{L}\, W_2(q\widehat{P}_{\mathsf{ODE}}^{(N_{\mathsf{pred}})}, q_{t_0+T_{\mathsf{pred}}}) + \varepsilon_{\mathsf{sc}}/\sqrt{L} + \sqrt{Ldh_{\mathsf{corr}}}\,. \quad (17)$$

For the Wasserstein term, Lemma 5 yields

$$W_2(q_{t_0}\widehat{P}_{\mathsf{ODE}}^{(N_{\mathsf{pred}})}, q_{t_0+T_{\mathsf{pred}}}) = W_2(q_{t_0}\widehat{P}_{\mathsf{ODE}}^{(N_{\mathsf{pred}})}, q_{t_0}P_{\mathsf{ODE}}^{(N_{\mathsf{pred}})}) \lesssim \sqrt{Ld}\, h_{\mathsf{pred}} + \frac{\varepsilon_{\mathsf{sc}}}{L}\,.$$

Combining these bounds yields the result for the overdamped corrector. For the underdamped corrector, we modify (17) by replacing the use of Theorem 4 with Theorem 5. □

We also need the following lemma on the convergence of the OU process.

**Lemma 13.** *Let $(q_t^{\rightarrow})_{t\geq 0}$ denote the marginal law of the OU process started at $q_0^{\rightarrow} = q_\star$. Then, for all $T \gtrsim 1$, it holds that*

$$\mathsf{TV}(q_T^{\rightarrow}, \gamma^d) \lesssim (\sqrt{d} + \mathfrak{m}_2)\exp(-T)\,.$$

*Proof.* This follows from [CLL23, Lemma C.4]. Alternatively, using the short-time regularization result of [BGL01, Lemma 4.2] for time $t_0 \asymp 1$ and the Wasserstein contraction of the OU process,

$$\mathsf{TV}(q_T^{\rightarrow}, \gamma^d) \lesssim \sqrt{\mathsf{KL}(q_T^{\rightarrow} \,\|\, \gamma^d)} \lesssim \frac{W_2(q_{T-t_0}^{\rightarrow}, \gamma^d)}{\sqrt{t_0}} \leq \exp(-(T-t_0))\, W_2(q_\star, \gamma^d)\,.$$

The result follows from $W_2(q_\star, \gamma^d) \leq W_2(q_\star, \delta_0) + W_2(\delta_0, \gamma^d) \leq \mathfrak{m}_2 + \sqrt{d}$. □

We now prove our main theorems.

*Proof of Theorems 2 and 3.* For $t \in [0, T]$, let $p_t := \mathrm{law}(\widehat{x}_t)$. From Lemma 13,

$$\mathsf{TV}(p_0, q_0) = \mathsf{TV}(q_T^{\rightarrow}, \gamma^d) \lesssim (\sqrt{d} + \mathfrak{m}_2)\exp(-T)\,.$$

We divide our analysis according to the two stages of the algorithm. In the first stage, after iterating Lemma 12 for $N_0 \asymp LT$ steps,

$$\mathsf{TV}(p_{T-h_{\mathsf{pred}}}, q_{T-h_{\mathsf{pred}}}) \leq \mathsf{TV}(p_0, q_0) + O\Big(L\sqrt{d}\, h_{\mathsf{pred}} + \sqrt{Ld}\, h_{\mathsf{corr}}^{\mathfrak{p}} + \frac{\varepsilon_{\mathsf{sc}}}{\sqrt{L}}\Big) \times N_0$$

$$\lesssim (\sqrt{d} + \mathfrak{m}_2)\exp(-T) + L^2 T d^{1/2} h_{\mathsf{pred}} + L^{3/2} T d^{1/2} h_{\mathsf{corr}}^{\mathfrak{p}} + L^{1/2} T \varepsilon_{\mathsf{sc}}$$

where $\mathfrak{p} = \frac{1}{2}$ if we use the overdamped corrector and $\mathfrak{p} = 1$ if we use the underdamped corrector. Applying the second part of Lemma 12 for the second stage of the algorithm, we then conclude that

$$\mathsf{TV}(p_{T-\delta}, q_{T-\delta}) \lesssim (\sqrt{d} + \mathfrak{m}_2)\exp(-T) + L^2 T d^{1/2} h_{\mathsf{pred}} + L^{3/2} T d^{1/2} h_{\mathsf{corr}}^{\mathfrak{p}} + L^{1/2} T \varepsilon_{\mathsf{sc}}\,.$$

Finally, we note that if we take $\delta \asymp \frac{\varepsilon^2}{L^2\,(d\vee\mathfrak{m}_2^2)}$, then by [LLT23, Lemma 6.4], $\mathsf{TV}(q_{T-\delta}, q_T) \leq \varepsilon$; a triangle inequality thus finishes the proof. □

*Remark* 4. Alternatively, instead of taking geometrically decreasing step sizes and employing early stopping, we could split the algorithm into two stages: for time $t < T - h_{\mathsf{pred}}$, we take constant step size $h_{\mathsf{pred}}$, and for time $t > T - h_{\mathsf{pred}}$, we use a smaller constant step size $h'$ as required if working with the original score perturbation lemma (see Remark 2).

# F  Numerical experiments

In this section, we provide preliminary numerical experiments to illustrate our theory. We implement DPUM on a toy model that is not log-concave (mixture of Gaussians).

**Setup.** The target distribution is a mixture of five Gaussians in dimension 5. On Figures 1 and 2, the red stars represent the means of the Gaussians and the red ellipses around the stars represent the variances of the Gaussians. We start by sampling 500 independent points (in blue) from a standard Gaussian. Then, we run DPUM from the blue dots over 300 iterations and plot the two first coordinates of the dots at iterations 0, 100, 200 and 300. This is a low-dimensional toy example so it does not illustrate our theory, rather we include it as a simple sanity check.

**Parameters.** We use a closed form formula for the score along the forward process. In other words, the score estimation error is equal to zero. The step size of the predictor is $0.01$ and the step size of the corrector is $0.001$. The corrector consists in 3 steps of the underdamped Langevin algorithm. In the latter algorithm, we initialize the velocity as a centered Gaussian random variable with standard deviation $0.001$ and set the parameter $\gamma$ to $0.01$.

**Observations.** We observe the expected behavior: although the target distribution is highly non-log-concave, DPUM is able to provide samples from a distribution that is close to the target distribution. In particular, the initial Gaussian distribution splits in clusters that will fit each component of the target mixture of Gaussians. Recall that we experiment without score error but with discretization error: our numerical results illustrate the common wisdom that score knowledge along the forward process can replace convexity assumptions. In particular, we observe that even isolated, low probability components of the Gaussian mixture, are recovered by DPUM.

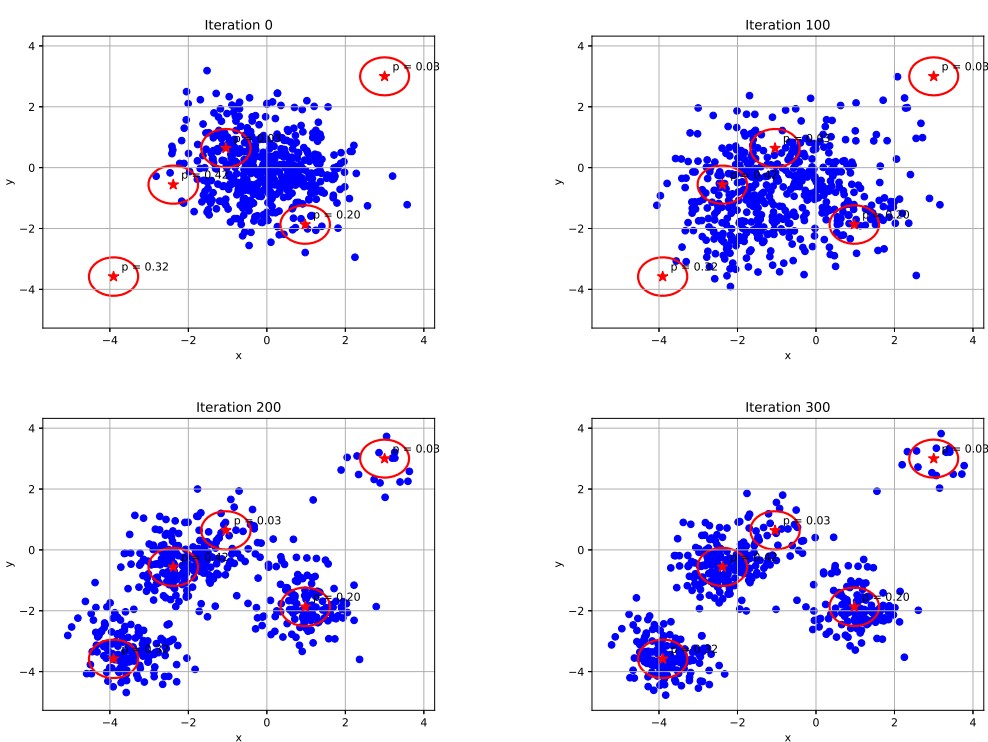

Figure 1: A realization of DPUM for a mixture of Gaussians.

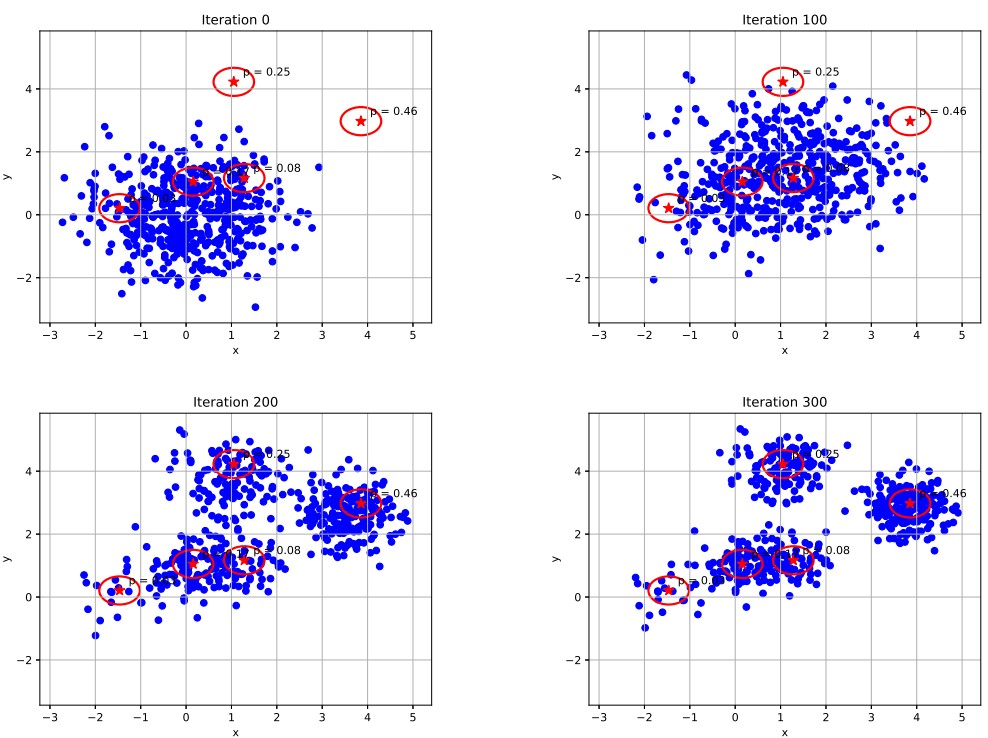

Figure 2: A realization of DPUM for another mixture of Gaussians.

