# OpenReview forum: "The probability flow ODE is provably fast"
_NeurIPS.cc/2023/Conference — NeurIPS 2023 poster_

### Official Review · Reviewer_bw63 · 2023-07-02

**Soundness:** 3 good
**Presentation:** 3 good
**Contribution:** 3 good
**Rating:** 5
**Confidence:** 3

**Summary:**

In this work, authors propose improved solution scheme with the specially chosen corrector step for the the score-based generative models. Within the denoising diffusion probabilistic modeling (DDPM) and the underdamped Langevin diffusion approach authors obtain better dimension dependence within certain restrictions on the smoothness of data distribution (Algorithm DPUM) than prior works.

**Strengths:**

1. Score-based generative models today are of great interest to the scientific community, while the theoretical establishment of their capabilities and limitations is a powerful impetus for the further development of new effective methods. The extensive analysis carried out in this work is certainly relevant and of direct interest to researchers.

2. The work is well structured, written in a great style and easy to read.

3. The provided approach with the polynomial-time convergence guarantees for the probability flow ODE seems to be essential if it will be successfully implemented in practice and applied to the real-world data.

**Weaknesses:**

Given the high level of theoretical consideration carried out in the work, I am confused by such a weak study of the experimental part:

1. It seems that a (at least brief) description of the performed numerical experiments should be included as a section of the main text (in the presented text, numerical experiments are contained exclusively in the appendix).

2. I do not quite understand the conclusions that can be drawn from the only simple numerical experiment presented in the appendix for artificially generated data from a mixture of Gaussians. Does the proposed theoretical estimate of the convergence of the method hold in this case? How to interpret a word "approximately" in [Line 721, appendix]: "DPUM is able to provide samples (approximately) from the target distribution"? Can any quantitative conclusions be drawn from the experiment? Is it possible to do some form of convergence analysis and make a comparison with the baseline?

3. Can numerical experiments be done on real-world datasets using a neural network ([Line 130]: "the score estimates are produced via a deep neural network"; [Line 723, appendix]: "recall that we experiment without score error")?

**Questions:**

Please see the section "Weaknesses" above. I have only one more minor comment: [Line 123] for clarity, it may be worthwhile, in addition to the dimension of the noise, to also indicate the dimension of the x variable.

**Limitations:**

The proposed approach is based on the assumption that the score functions along the forward process are Lipschitz continuous, and it does not cover non-smooth distributions. However, the authors note that this limitation can potentially be circumvented by using the early stopping technique.

---

> ### Author Rebuttal · Authors · 2023-08-08
>
> Thanks for your positive comments.
>
> **I am confused by such a weak study of the experimental part**
>
> Please refer to our answer to all reviewers above.
>
>  As mentioned in the paper: “Although we provide preliminary numerical experiments in the Appendix, we leave it as a question for future work to determine whether the theoretical benefits of the underdamped Langevin corrector are also borne out in practice”.
>
> The experiment is purely illustrative: we believe experimenting should always start with the simplest relevant model, that’s why we used a mixture of Gaussians as a toy model.
>
>
> **It seems that a (at least brief) description of the performed numerical experiments should be included as a section of the main text (in the presented text, numerical experiments are contained exclusively in the appendix).**
>
> The experiment we provided is purely illustrative, that’s why we chose to put it in the appendix, and we might remove them from the next version of the paper if it confuses the reader about the theoretical nature of our paper.
>
> **I do not quite understand the conclusions that can be drawn from the only simple numerical experiment presented in the appendix for artificially generated data from a mixture of Gaussians.**
>
> We experimented DPUM on a toy model so the only conclusion one can draw is that DPUM works on this toy model. But this experiment is not our main contribution. Our theory, which is general and not specific to a toy model, is our main contribution.
>
>
> **Does the proposed theoretical estimate of the convergence of the method hold in this case?**
>
> Yes, because in this toy model the score can be obtained in closed form as mentioned in the experiment section “We use a closed form formula for the score along the forward process.” Therefore, the theoretical estimate trivially holds. We will clarify this point.
>
> **How to interpret a word "approximately" in [Line 721, appendix]: "DPUM is able to provide samples (approximately) from the target distribution"?**
>
> The word “(approximately)” refers to the fact that the output of the algo DPUM is a distribution that is close (but not equal) to the target distribution. Our theory quantifies the distance between the two distributions. We will clarify this point.
>
> **Can any quantitative conclusions be drawn from the experiment? Is it possible to do some form of convergence analysis and make a comparison with the baseline?**
>
> Absolutely. Our theory applies to this toy model.
>
> In particular, we can specify our Theorem3 to the toy model and explicit the values of the constants. Is it what you are asking for? If yes, we can add these calculations to the experimental section. By the way, these calculations were needed to run our simulations (whose code is provided in the Supplementary material).
>
> Besides, could you precise what you mean by “comparison to the baseline” please?
>
>
> **Can numerical experiments be done on real-world datasets using a neural network**
>
> Please refer to our answer to all reviewers above.
>
> **The proposed approach is based on the assumption that the score functions along the forward process are Lipschitz continuous, and it does not cover non-smooth distributions. However, the authors note that this limitation can potentially be circumvented by using the early stopping technique.**
>
>
> There is a routine "early stopping" technique to overcome the smoothness assumption, see [Che+23; LLT23]. We can apply this technique here: we can add a corollary that covers nonsmoothess if you believe that it is important. All the guarantees will remain polynomial in every parameters of interest.
>
> In order to keep the paper simple, we chose to explain the early stopping argument (which is quite simple) without giving the final complexity result.
>
>
> **We kindly ask you to evaluate our paper with the lens of a theoretician, i.e., in light of our actual contributions (towards theoretical understanding of diffusion models). If you believe that our theoretical contribution is enough for publication please consider raising your score.**
>
>
>
>
> [Che+23]: S. Chen, S. Chewi, J. Li, Y. Li, A. Salim, and A. R. Zhang. “Sampling is as easy as learning the score: theory for diffusion models with minimal data assumptions”, ICLR 2023
>
> [LLT23]: H. Lee, J. Lu, and Y. Tan. “Convergence of score-based generative modeling for general data distributions”, ALT 2023

---

> > ### Comment · Reviewer_bw63 · 2023-08-10
> >
> > Thank you for your response. First of all, I would like to clarify the following point. In your response you stated: `We experimented DPUM on a toy model so the only conclusion one can draw is that DPUM works on this toy model. But this experiment is not our main contribution.`
> >
> > However, in section "1.1 Our contributions" you specify: ...
> > - We propose an algorithm based on the probability flow ODE with underdamped Langevin corrector (DPUM..
> > - We provide the first convergence guarantees for DPUM. These convergence guarantees show improvement over (i) the complexity of DPOM ...
> > - We provide preliminary numerical experiments in a toy example showing that DPUM can sample from a highly non log-concave distribution (see Appendix)...
> >
> > It seems to me that the reader may get the feeling that your numerical experiment confirms your theoretical estimates for convergence. In this context, I would like to clarify whether it is possible to see confirmation of your main result from the experiment you conducted? If not, why not? [By baseline in my review, I meant DPOM].

---

> > > ### Author Response · Authors · 2023-08-11
> > > **Thanks for engaging with us**
> > >
> > > Thanks for your reply.
> > >
> > > Exactly. The experiment does NOT illustrate the theory because our theorems holds for large dimension, whereas the experiment is done in low dimension. In particular, we did not see a difference between DPOM and DPUM in our low dimensional experiment (and we did not expect to see a difference).
> > >
> > > Therefore, the experiment and the theory are somehow orthogonal. So, one may ask why to include the experiment. The reason is simply because DPUM is a new algorithm that we propose. So, we thought that it would be interesting to try it on the simplest relevant case, which is in our opinion a low dimensional mixture of Gaussian.
> > >
> > > We will clarify that the experiment is not our main contribution and that it does not illustrate our theory (which holds when d is large).
> > >
> > > Finally, the reason why we do not increase the dimension in the experiment is because we run into instability and memory issues (one can try by setting a large d in the code we provided in the Supplementary).

---

> > > > ### Comment · Reviewer_bw63 · 2023-08-11
> > > >
> > > > Thank you.
> > > >
> > > > 1. Statements `DPUM is a new algorithm that we propose` and `we run into instability and memory issues` in your answer make me agree with Reviewer L7v8 ("I think for a paper about diffusion models, theoretical results and convincing experiments are both very important. Otherwise, there would be too many papers with beautiful theories that can hardly be used in real applications"). On the other hand, indeed, there is a significant number of successful works on this topic that contain only theoretical constructions, and your request to evaluate only your theoretical results, in general, sounds (to my regret) logical.
> > > >
> > > > 2. You chose a somewhat aggressive style of responding to the comments of the Reviewers, and it seems to me that this is slightly inappropriate, including, for example, such a phrase for one of the Reviewers as "...you rated your confidence level very low compared to the other reviewers. We kindly ask you to raise your confidence level...". On the other hand, you actively and purposefully defend your creation, and this, in general, causes respect.
> > > >
> > > > Since the comments in my original review were specifically about the experimental part, taking into account the above, I raise my final score from 4 to 5. I wish you good luck and further scientific success!

---

> > > > > ### Author Response · Authors · 2023-08-11
> > > > > **Thanks for raising the score.**
> > > > >
> > > > > Thank you for your positive comments.
> > > > >
> > > > > We are sorry if you find our style aggressive.
> > > > >
> > > > > To be honest we wondered if we should write the sentence you refer to. But we believe that the low confidence level of the review you refer to does not match the level of details in the comments and questions of that review *regarding our theorems*, which are our main contributions.
> > > > >
> > > > > Anyways, we also wish you good luck and scientific success in the future.

---

> > > > > > ### Author Response · Authors · 2023-08-11
> > > > > > **Last clarification**
> > > > > >
> > > > > > We want to clarify a last point:
> > > > > >
> > > > > > The instability issue we ran into in high dimension **are not specific to DPUM**, we ran into the same issues with DPOM, without corrector, and with SDE based diffusion model (see the code in the Supplementary where all these methods are implemented). We believe that this is because we have to fine-tune the hyperparameters for each of these methods.
> > > > > >
> > > > > > Best

---

> > > > > > > ### Author Response · Authors · 2023-08-18
> > > > > > > **Simulation in higher dimension**
> > > > > > >
> > > > > > > **We resolved the instability issue.**
> > > > > > >
> > > > > > > It was due to the calculation of the score of the mixture of Gaussian: the density of the mixture of Gaussian in high dimension can be so low that it was treated by zero by Python. The score is needed by DPUM and other concurrent methods, that's why we could not experiment in dimension 500 for example.
> > > > > > >
> > > > > > > We resolved the instability by multiplying the density by a large constant (which corresponds to the exponential of the entropy of the Gaussian components of the mixture). That does not change the value of the score (since the score involves calculating the gradient). The code is given at the end of this Comment.
> > > > > > >
> > > > > > >
> > > > > > > **Observation**: Our paper contains simulations of 500 particles in dimension 5 along 300 iterations. We see that in dimension 180, our algorithm's output at iteration 1800 is equivalent to the iteration 300 in dimension 5. In other words, by multiplying the dimension by 36, the algorithm is 6 times slower. Since $6 = \sqrt{36}$, this simulation suggests the scaling $O(\sqrt{d})$ for DPUM, which is mathematically proven in the paper.
> > > > > > >
> > > > > > >
> > > > > > > *We hope that this improves your opinion of our paper.*
> > > > > > >
> > > > > > >
> > > > > > > **Code**: in the Python file in the supplementary, replace the line
> > > > > > >
> > > > > > > densities[:, i] = torch.distributions.MultivariateNormal(mean, cov).log_prob(X).exp().to(X.device)
> > > > > > >
> > > > > > > by the lines
> > > > > > >
> > > > > > > dim = cov.shape[0]
> > > > > > >
> > > > > > > determinant = torch.det(cov)
> > > > > > >
> > > > > > > constant_term = torch.tensor(2 * torch.pi * torch.e, dtype=torch.float)
> > > > > > >
> > > > > > > entropy = 0.5 * (dim * torch.log(constant_term) + torch.log(determinant))
> > > > > > >
> > > > > > > A = torch.distributions.MultivariateNormal(mean, cov).log_prob(X) + entropy
> > > > > > >
> > > > > > > densities[:, i] = A.exp().to(X.device)

---

### Official Review · Reviewer_L7v8 · 2023-07-04

**Soundness:** 2 fair
**Presentation:** 3 good
**Contribution:** 2 fair
**Rating:** 4
**Confidence:** 4

**Summary:**

This paper provides polynomial-time convergence guarantees for the probability flow ODE implementation. This paper obtains a better better dimension dependence than prior works on DDPM. However, there is no experiments to prove the effectiveness of the proposed methods.

**Strengths:**

1. The paper is well written.
2. The theoretical results are solid.

**Weaknesses:**

1. No experiments are performed to prove the proposed method. I think it is necessary to evaluate the method of diffusion models, such as Stable Diffusion or DeepFloyd. I admit that the theoretical results presented in this paper might be important, but that is not the reason of the lack of experiments.
2. The proposed method has an extra corrector step, which further introduces extra computational costs.
3. Some comparisons with previous sampling methods (DPM-Solver++, DEIS, etc) are missing.

**Questions:**

Please see the weaknesses.

**Limitations:**

The limitations have been fully discussed.

---

> ### Author Rebuttal · Authors · 2023-08-08
>
> Thank you for the review.
>
>
> **No experiments are performed to prove the proposed method.**
>
>
> We experimented our main algorithm DPUM in the appendix on a toy model, and **the Python code is included in the Supplementary material.** The other algorithm we analyze in the paper, DPOM, was already experimented in [Son+21a, Section 4.2].
>
> **I think it is necessary to evaluate the method of diffusion models, such as Stable Diffusion or DeepFloyd.**
>
> See our comment to all reviewers above. We strongly disagree with this statement. Nobody asked the Stable Diffusion paper or the DeepFloyd paper to provide extensive theory.  Similarly, we believe that papers with solid theory and with small / without experiments can be published as well.
>
> **The proposed method has an extra corrector step, which further introduces extra computational costs.**
>
> **This is wrong.** The complexity we give is the **TOTAL** complexity including the number of steps in the subroutine.
>
> Besides, the use of correctors has shown superiority in practice, see [Son+21a, Section 4.2] among others.
>
> **Some comparisons with previous sampling methods (DPM-Solver++, DEIS, etc) are missing.**
>
> **Comparison is impossible** because all these other samplers are lacking theory. There is an absolute need for more theory papers, similar to ours, in the field of diffusion models.
>
>
>
> **We kindly ask you to evaluate our paper with the lens of a theoretician, i.e., in light of our actual contributions (towards theoretical understanding of diffusion models). If you believe that our theoretical contribution is enough for publication please consider raising your score.**
>
>
>
>
>
>
> [Son+21a]: Y. Song, C. Durkan, I. Murray, and S. Ermon. “Maximum likelihood training of score based diffusion models”, NeurIPS 2021

---

> > ### Comment · Reviewer_L7v8 · 2023-08-11
> >
> > Thanks for the authors' response. I think for a paper about diffusion models, theoretical results and convincing experiments are both very important. Otherwise, there would be too many papers with beautiful theories that can hardly be used in real applications. Since the authors have addressed most of my concerns, I would raise my score to borderline reject.

---

> > > ### Author Response · Authors · 2023-08-11
> > > **Thanks for raising the score.**
> > >
> > > We respect your opinion but we still believe that diffusion model is a field where there is a disequilibrium between the amount of applied works compared to the amount of theory works. The field is lacking theory papers that provides mathematical understanding.
> > >
> > > **Expecting from a theory paper like ours to provide large scale experiments fuels this imbalance between theory and practice in the field by setting the bar too high for theoretical papers.**
> > >
> > > Recall that we do not expect applied papers to provide extensive theory (and that's the right thing to do!) so it feels unfair to expect from us extensive experiments.
> > >
> > > Thanks for your consideration.

---

> > > > ### Comment · Reviewer_Q98a · 2023-08-11
> > > > **I personally agree.**
> > > >
> > > > A theoretical paper should be mainly evaluated by the theoretical novelty, application potential, and soundness.

---

> > > > > ### Author Response · Authors · 2023-08-18
> > > > > **Simulation in higher dimension**
> > > > >
> > > > > We scaled our toy experiment to higher dimension. We know this is not exactly what you were looking for, but we still hope that this improves your opinion of our paper.
> > > > >
> > > > > Recall that our paper contains simulations of 500 particles in dimension 5 along 300 iterations. We have modified the code to run simulations in higher dimension (e.g., d = 500).
> > > > >
> > > > > In particular, we observe that in dimension 180, our algorithm's output at iteration 1800 is equivalent to the iteration 300 in dimension 5. In other words, by multiplying the dimension by 36, the algorithm is 6 times slower. Since $6 = \sqrt{36}$, this simulation suggests the scaling $O(\sqrt{d})$ for DPUM, which is mathematically proven in the paper.

---

### Official Review · Reviewer_Q98a · 2023-07-04

**Soundness:** 3 good
**Presentation:** 3 good
**Contribution:** 3 good
**Rating:** 6
**Confidence:** 4

**Summary:**

The authors proposed the convergence guarantees for the probability flow ODE implementation. Through the use of a specially chosen corrector step based on the underdamped Langevin diffusion, a better dimension dependence of $O(\sqrt{d})$ than DDPM $O(d)$ is achieved.

**Strengths:**

1. the theoretical comparison between probability flow ODE and DDPM is less studied in the community. In this sense, analyzing the advantage of the ODE-based method brings new insight to the community.

2. solid and clean derivations.

**Weaknesses:**

1. the term of fast looks a bit confusing to me. It is fast in inference or training? they have different meanings. As provided in the ranking: https://paperswithcode.com/sota/image-generation-on-cifar-10, **flow-based methods seem to perform quite mediocre in standard datasets**. So at least the claim of fast flow-ODE contradicts the fact of fast training in practice. Clarification is suggested.

2. Flow-based methods are known to work well in **low dimensions** and I am even suspecting it suffers from the curse of dimensionality. As such. I appreciate the authors' solid math techniques, but I personally believe that the advantage of the well-known smaller discretization error $O(\sqrt{d})$ v.s. $O(d)$ becomes **less interesting if it mainly adapts to low-dimensional problems** (see Q1 below). the authors can correct me if I am wrong.

3. The authors should point out the **limitations of probability-flow ODE methods** compared to DDPM so as not to confuse readers.



**Questions:**

Q1. Do you know any popular real-world examples that probability-flow ODE works well in ultra-high dimensions in practice?

Q2. Growall inequality does suffer from the exponential blowup.
Q2.a) The authors propose to use the geometric contraction of the corrector (diffusion) step to control the exponential blowup. Is that right?
Q2.b) Does it mean we can do lots of ODE steps but with only a few diffusion steps, the exponential blowup is still controlled in terms of complexity (ignore the constant) due to the contraction (closer to 1, e.g. alpha=0.999999999 also works, although the error / (1-alpha) is large)?
Q2.c) Or do you need to maintain at least some amount of diffusion steps to make sure the bound hold?

Minor: a relative work is missing and a short comment on this work would be appreciated: Provably Convergent Schrodinger Bridge with Applications to Probabilistic Time Series Imputation. ICML'23.

---

> ### Author Rebuttal · Authors · 2023-08-09
>
> Thank you for your positive review.
>
> **The term of fast looks a bit confusing to me. It is fast in inference or training?**
>
> Our theory shows that sampling is fast, therefore the term fast refers to the inference time, not the training time. We will clarify this point in the introduction.
>
> **Flow-based methods are known to work well in low dimensions and I am even suspecting it suffers from the curse of dimensionality. As such, I appreciate the authors' solid math techniques, but I personally believe that the advantage of the well-known smaller discretization error becomes less interesting if it mainly adapts to low-dimensional problems (see Q1 below). The authors can correct me if I am wrong.**
>
> Our analysis separates the study of score estimation from the sampling (i.e., inference) complexity, and we only study the latter. Specifically, we show that **if score estimation is possible with low error, then the sampling complexity does NOT suffer from the curse of dimensionality.**
>
> That being said, there may be situations in which score estimation incurs the curse of dimensionality (in general this is true). However, there may be other situations in which score estimation does not incur the curse. For example, this may be the case when the data has some intrinsic low-dimensional structure and the neural network training is able to adapt to it. **Regardless of what situation we are in, our results hold independently.**
>
> Finally, slow training in high dimension is a *practical* observation, whereas our paper *mathematically proves* faster inference. Hence, we disagree with your assessment that our results are only interesting in low dimensions. It depends on the context.
>
> **The authors should point out the limitations of probability-flow ODE methods compared to DDPM so as not to confuse readers.**
>
> Thanks for the suggestion. We will mention this limitation (curse of dimensionality in training phase observed in practice) in our paper.
>
>
> **Q1. Do you know any popular real-world examples that probability-flow ODE works well in ultra-high dimensions in practice?**
>
> For ultra-high dimension, like generating videos, we do not know and we believe it is still an open problem.
>
>
> **Q2. Growall inequality does suffer from the exponential blowup.**
>
> **Q2.a) The authors propose to use the geometric contraction of the corrector (diffusion) step to control the exponential blowup. Is that right?**
>
> To clarify, the corrector does NOT have a geometric contraction; in fact, the corrector step also suffers from an exponential blowup in the Wasserstein distance. Technically, the role of the corrector step is to *add stochasticity*, which allows us to transfer Wasserstein error to TV error which accumulates linearly, rather than exponentially over iterations. Intuitively, with stochasticity, we can argue that the probability that the reverse diffusion and the algorithm diverge is very small; but without any stochasticity, the analysis must be “worst-case” and hence leads to poor bounds.
>
>
> **Q2.b) Does it mean we can do lots of ODE steps but with only a few diffusion steps, the exponential blowup is still controlled in terms of complexity (ignore the constant) due to the contraction (closer to 1, e.g. alpha=0.999999999 also works, although the error / (1-alpha) is large)?**
>
> This is answered by the previous point, namely that even with the corrector step, there is no contraction.
>
> **Q2.c)  Or do you need to maintain at least some amount of diffusion steps to make sure the bound hold?**
>
> Our bounds do rely on taking enough diffusion steps to ensure that there is enough added stochasticity.
>
>
> **MINOR**: Thank you for pointing out this work, we will add it to the literature review.
>
>
>
> *We hope we answered your concerns and we would be happy if you could raise your score if you are satisfied with our answers.*

---

> > ### Comment · Reviewer_Q98a · 2023-08-11
> > **ask some details**
> >
> > 1. I probably missed something in the literature. Could you share some standard references on when Langevin leads a linearly-increased error in TV?
> >
> > 2. Could you show me in your proof where we require enough diffusion steps to yield this linear property?

---

> > > ### Author Response · Authors · 2023-08-11
> > > **Thanks for the questions**
> > >
> > > Thanks again for commenting and questioning our theorems.
> > >
> > > 1. The analysis in TV which leads to error growing linearly in time is not found in the literature; rather it is a **novelty of our work**. Intuitively this stems from different ways of measuring distances between probability distributions. The Wasserstein distance measures distances between random variables, and since our dynamics are not contractive, then a naive coupling of the algorithm iterates with the ground truth lead to errors being compounded over time, leading to an exponential blow-up. But when we work in total variation distance, we work with probabilities of certain events. At each iteration, there is some probability of the "bad event" occurring, in which the algorithm iterate has strayed away from the ground truth. But by the union bound we can sum up the probabilities of these bad events across the iterations, leading to an error bound that only grows linearly with the number of iterations. This is the main reason why we develop our framework in which the added stochasticity of the diffusion leads to a Wasserstein-to-TV regularization effect, and then we carry out our analysis in TV.
> > >
> > > 2. As mentioned above, the key property of the diffusion is Wasserstein-to-TV regularization, which is captured in Thms. 4 and 5 in the Appendix. These theorems show that after running the corrector, we can bound the TV error in terms of the Wasserstein error. However, the bounds depend on the amount of continuous time for which we run the corrector. For our analysis, we run the corrector for sufficiently long in order for these bounds to give good results. **To see the linear accumulation of error in TV, see the proof of Thms. 2 and 3 in the Appendix; in particular, in the second display equation, the error terms scale linearly with T, the total time that we run the process.**

---

> > > > ### Comment · Reviewer_Q98a · 2023-08-12
> > > > **Thanks for the response**
> > > >
> > > > I think the authors have addressed my concerns on the theoretical side. Their response has bolstered my confidence, and I genuinely appreciate their efforts.
> > > >
> > > > In terms of practical experimentation, I hold the view that real-world experiments, such as CIFAR-10, might not be essential for this solid theoretical diffusion paper. Nonetheless, considering the paper's emphasis on exhibiting enhanced dimension dependence, it would be prudent to consider employing clever high-dimensional synthetic simulations (which would help increase my current score). Drawing inspiration from [1], where a sampler was employed with a dimensionality of 1000, could serve as an apt illustration of the paper's strengths.
> > > >
> > > > With these thoughts in mind, I am inclined to maintain my current score of 6.
> > > >
> > > > [1] Dimensionally Tight Bounds for Second-Order Hamiltonian Monte Carlo.

---

> > > > > ### Author Response · Authors · 2023-08-18
> > > > > **Simulation in higher dimension**
> > > > >
> > > > > Thanks for the suggestion and for your support.
> > > > >
> > > > > Our paper contains simulations of 500 particles in dimension 5 along 300 iterations. We have modified the code to run simulations in higher dimension (e.g., d = 500).
> > > > >
> > > > >
> > > > > In particular, we observe that in dimension 180, our algorithm's output at iteration 1800 is equivalent to the iteration 300 in dimension 5. In other words, by multiplying the dimension by 36, the algorithm is 6 times slower. Since $6 = \sqrt{36}$, this simulation suggests the scaling $O(\sqrt{d})$ for DPUM, which is mathematically proven in the paper.
> > > > >
> > > > >
> > > > > **We hope that this improves your opinion of our paper.**

---

### Official Review · Reviewer_tQnV · 2023-07-05

**Soundness:** 4 excellent
**Presentation:** 4 excellent
**Contribution:** 3 good
**Rating:** 8
**Confidence:** 3

**Summary:**

The current study offers a polynomial-time total variation convergence guarantee for score-based generative modeling, given that the generative process is a probabilistic Ordinary Differential Equation (ODE). This differs from previous work, which typically involves a reverse Stochastic Differential Equation (SDE).  To the best of my knowledge, this is the first paper to present such results for an ODE-implemented score-based generative model targeting a general distribution without assuming log-concavity. The proof technique includes analyzing the discretization error of both the ODE and SDE, and employs the Score Perturbation Lemma to limit the error introduced by using estimated score functions.

It's important to note that the convergence guarantee is provided for a modified version of the ODE-implemented score-based generative model. The authors have added an additional stochastic correction step to the deterministic dynamics, which is an unconventional approach.

**Strengths:**

The results provided in the work is novel and important in my perspective. As mentioned in the summary, this is the first convergence guarantee for an ODE-based, score-based generative model for general target distributions that I am aware of (this could, of course, be due to my limited familiarity with the relevant literature). While some may contend that the additional stochastic correction step diminishes the impact of the work (given that the primary challenge of the proof is that the ODE transformation is purely deterministic, making Total Variation/Kullback-Leibler control nearly impossible), the theoretical framework employed remains significantly distinct from its SDE counterparts.

The clarity of the writing and the structure of the proof are commendable. Despite the complexity of the paper, the proof is not difficult to follow, largely owing to a comprehensive preliminary introduction to the proof techniques. I particularly value the fact that the authors dedicated a significant portion of the text to discussing the limitations of some existing proof concepts, and made comparisons with similar intermediate results from the literature. This approach further underscores the contributions of this work.



**Weaknesses:**

The inclusion of an additional stochastic corrector in the generative process is somewhat unsatisfying. To the best of my knowledge regarding score-based generative modeling via ODE, this correction step is not typically included in practice. One aspect that I believe is not clearly addressed in the current results is the comparison between the corrected and uncorrected generative processes. Ideally, there would be a result showing that the corrected process converges to the target faster.
While it is mentioned in the paper that we cannot expect such a comparison in terms of Total Variation (TV) or Kullback-Leibler (KL) divergence - as without the stochastic correction, the TV/KL bound would be vacuous - could we not show such a comparison in terms of weak distance or even Wasserstein distance?
(I've also raised some questions regarding the corrector in the Questions section.)

Another minor shortcoming pertains to the toy numerical experiments presented in the study. It would have been more desirable if the authors had evaluated Algorithms 1 & 2 on actual image generation tasks and compared the performance of the corrected and uncorrected versions in these scenarios. While I admit it might seem somewhat unfair to expect this from a theory-oriented paper, such empirical evidence would undoubtedly render the study more comprehensive and impactful.


**Questions:**

* Step 5 for Alg 1 and 2: Can authors provide some intuition on why geometric decreasing stepsize is needed for the prediction at the end given that correction step is used eventually? (I understand from the proof that when $t>1/L$, decreasing step size is required to control the ODE discretization error + approximated score error, but cannot form a practical intuition on this.)

* In practice, which one of  DPOM and DPUM is preferred? Or how does one choose the algorithm for practical applications?

* Is it possible to get rid of the correction step for the convergence if the ODE discretization error is reduce by using more sphisticated numerical integrator (e.g., high order sympletic integrator)?  Based on my superfluous understanding of the proof, the answer is probably NO since the key of the corrector is to obtain TV bounds based on Wasserstein bounds, intead of "correcting" the introduced error during the ODE simulation. This leads to my next question.

* If intead of TV distance, all we care is just wasserstein distance, can we get rid of the correction step? Or is the correction step beneficial for a faster convergence?



**Limitations:**

The author did note limitations or boundaries of the adopted techniques through out the proof (e.g., line 565 in Appendix C), and raised practical questions base on the theoretical framework.

---

> ### Author Rebuttal · Authors · 2023-08-09
>
> Thank you very much for the generous review.
>
> **The inclusion of an additional stochastic corrector in the generative process is somewhat unsatisfying [...]**
>
> Regarding the use of the corrector, we are not the first ones to propose this; in fact, it was shown to achieve superior performances in practice in [Son+21a, Section 4.2] and several subsequent works. Nevertheless, we acknowledge that we could not obtain polynomial-time guarantees without this corrector step, and it is an important open question to determine whether the corrector step is necessary. Note however that in Wasserstein distance, we can easily obtain guarantees for both the corrected and uncorrected algorithms which scale exponentially in the problem parameters (but for this the corrector is useless).
>
>
> **Another minor shortcoming pertains to the toy numerical experiments presented in the study. It would have been more desirable if the authors had evaluated Algorithms 1 & 2 on actual image generation tasks and compared the performance of the corrected and uncorrected versions in these scenarios. While I admit it might seem somewhat unfair to expect this from a theory-oriented paper, such empirical evidence would undoubtedly render the study more comprehensive and impactful.**
>
> Thank you for acknowledging that it might seem somewhat unfair to expect this from a theory-oriented paper (see also our answer to all reviewers). We agree that such experiments would make our paper even stronger, but we prefer to let this for future work.
>
> **Step 5 for Alg 1 and 2: Can authors provide some intuition on why geometric decreasing stepsize is needed for the prediction at the end given that correction step is used eventually?**
>
> In our analysis, we use geometrically decreasing step sizes because our improved score perturbation lemma degrades as time approaches 0. Alternatively, we could use a two-stage step size schedule (see Remark 4) in the Appendix. Intuitively, this suggests that the discretization needs to be more accurate near time 0 due to the poor smoothness of the score function; we are not sure if it is necessary though. We will add comments explaining this to our proof overview.
>
>
> **In practice, which one of DPOM and DPUM is preferred? Or how does one choose the algorithm for practical applications?**
>
> Our theoretical results suggest that DPUM is more effective especially in high dimension. But in reality, it also depends on the score estimation error, and more extensive experiments are needed to properly answer this question. At least, we did not see a difference between DPUM and DPOM on our low dimensional toy experiments (and we didn't expect to see a difference in low dimension). The Python code is provided in the Supplementary if you want to try.
>
>
> **Is it possible to get rid of the correction step for the convergence if the ODE discretization error is reduced by using more sophisticated numerical integrator (e.g., high order symplectic integrator)?**
>
> This is an interesting idea that deserves to be explored. For the moment, we do not know if it is possible. One issue we see is that higher-order integrators usually require higher-order smoothness assumptions to be effective, which we want to avoid.
>
> **Based on my superfluous understanding of the proof, the answer is probably NO since the key of the corrector is to obtain TV bounds based on Wasserstein bounds, instead of "correcting" the introduced error during the ODE simulation. This leads to my next question.**
>
> To be more precise, the corrector allows us to use the TV distance for which the error accumulates *linearly* over the number of iterations. In contrast, due to the lack of contractivity of the dynamics, the Wasserstein error tends to compound *exponentially* over the iterations. Using higher-order integrators to resolve this issue seems challenging.
>
> **If instead of TV distance, all we care is just wasserstein distance, can we get rid of the correction step? Or is the correction step beneficial for a faster convergence?**
>
> As we just said, the exponential accumulation of the error in Wasserstein distance is a problem and we are currently unable to get polynomial-time guarantees in Wasserstein distance (with or without corrector). In summary, the corrector is necessary for our proof to give polynomial, non-exponential, bounds.
>
>
>
> However, there is a restricted setting in which we can give polynomial bounds in Wasserstein distance and without corrector. For this to work, we need to use the early stopping technique used in [Che+23] and to assume the following:
>
> - Zero score estimation error
> - Zero discretization error.
>
> In this setting, the "algorithm" outputs the distribution obtained by running the Prob Flow ODE **exactly**, but from the standard Gaussian (instead of the end distribution of the forward process). We are able to show polynomial bounds for this "algorithm" in terms of the Wasserstein distance to the target distribution. But this is a restricted setting.
>
> -------------------------------------------------------------
>
> Dear Reviewer,
>
> Given your review, you are obviously among the reviewers who understood our work the best. However, you rated your confidence level very low compared to the other reviewers. We kindly ask you to raise your confidence level, by comparison to the others. And we would be happy if you could raise your score if you are satisfied with our answers.
>
>
>
>
> [Son+21a]: Y. Song, C. Durkan, I. Murray, and S. Ermon. “Maximum likelihood training of score based diffusion models”, NeurIPS 2021
>
> [Che+23]: S. Chen, S. Chewi, J. Li, Y. Li, A. Salim, and A. R. Zhang. “Sampling is as easy as learning the score: theory for diffusion models with minimal data assumptions”, ICLR 2023

---

> > ### Comment · Reviewer_tQnV · 2023-08-12
> > **Response to authors rebuttal**
> >
> > Thank you for answering my questions; I think your response has addressed my questions and clarifies further the key novelty of the presented proof technique. However, I decided to keep the score 8 for this work, which clearly means it's a strong theory paper. To merit a score above 9 in my evaluation criteria, a paper should excel both theoretically and empirically.
> >
> > Regarding my initial confidence score, I wish to clarify that it was not due to my lack of ability to understand the content, but to from my unfamiliarity with the current literatures on theories for ODE based diffussion models. So I felt less confident to assess the significance of the current results. Now after reading up some literature in this line of work, I'm more confident to my evaluation. In this regards, I raised my confidence score from 2 to 3.

---

> > > ### Author Response · Authors · 2023-08-18
> > > **Simulation in higher dimension**
> > >
> > > Thanks for your support and for updating the confidence level.
> > >
> > > FYI, we performed simulations in higher dimension.
> > >
> > > Our paper contains simulations of 500 particles in dimension 5 along 300 iterations. We have modified the code to run simulations in higher dimension (e.g., d = 500).
> > >
> > >
> > > In particular, we observe that in dimension 180, our algorithm's output at iteration 1800 is equivalent to the iteration 300 in dimension 5. In other words, by multiplying the dimension by 36, the algorithm is 6 times slower. Since $6 = \sqrt{36},$ this simulation suggests the scaling $O(\sqrt{d})$ for DPUM, which is mathematically proven in the paper.

---

### Official Review · Reviewer_rLPm · 2023-07-06

**Soundness:** 2 fair
**Presentation:** 2 fair
**Contribution:** 3 good
**Rating:** 5
**Confidence:** 3

**Summary:**

In this article the authors propose polynomial-time convergence guarantees for probability flow ODEs related to score based generative models.
Previous analyses on score based diffusion models have resorted to the use of SDEs and relied on properties of stochastic flows to alleviate error accumulation due to discretisation. Here the authors study the ODE counterparts of diffusion models.
Since the deterministic dynamics are not contractive without log-concave assumptions on the data distribution, and thereby
discretisation errors accumulate, the authors derive bounds for a modified process which requires interleaving deterministic integration steps (predictor) with stochastic (corrector) steps that smooths the reverse process, drawing inspiration from the purely stochastic framework of [4,9]. They propose two variants of the algorithm, one relying on overdamped Langevin dynamics with dimension dependence that scales as $\mathcal{O}(d)$ ,and one with underdamped Langevin dynamics that scales more favourably as $\mathcal{O}(\sqrt{d})$and thus has improved dimension dependence compared to previous approaches.

The topic of the paper is interesting and relevant for the audience of  NeurIPS for understanding denoising diffusion probabilistic models, however I find that the article could have been more accessible for the broader audience of NeurIPS, and importantly, I would expect that the authors would provide numerical evidence for their proposed guarantees.

**Strengths:**

- the authors provide improved convergence guarantees compared to existing approaches.
- they relax log-concavity assumptions existing in previous frameworks that are rather restrictive.

**Weaknesses:**

- The work is purely theoretical and the authors do not provide numerical evidence for their claims excluding a simple experiment in the supplement. I find it a bit odd that they provide a convergence guarantee and propose a scaling with system dimension, but nevertheless they don't explore the said scaling numerically for increasing system dimension. The numerical example they present in the appendix concerns only a single experiment of dimension $d=5$.

- the main theorem of the paper is formulated considering only linear dynamics (Ornstein-Uhlenbeck process). However according to the introduction of the
 paper a reader gets the impression that the provided results are relevant for general dynamics. I would propose to introduce this clarification both in the abstract and the introduction of the
paper, otherwise the statements may be considered misleading.


- the derivation of the probability flow ODE in Eq. 3 is a bit odd, and happens to be correct here because the considered dynamics is conservative. The whole point of [3]  first introduced the
formalism of ODE dynamics for diffusion that got later named prob. flow ODE by [4], was that there is no need for a stationary solution for the prob. flow ODE and no need to derive the dynamics wrt to the stationary solution.
For general systems one can derive the dynamics of the deterministic process by rewriting the Fokker-Planck equation in the form of a Liouville equation
as
\begin{align}
\partial_t p_t(\mathbf{x})& = - \nabla \cdot \left( f(\mathbf{x}) p_t(\mathbf{x}) - \frac{\sigma^2}{2} \nabla p_t(\mathbf{x})  \right)   \\\\
&=- \nabla \cdot \left[p_t(\mathbf{x})  \left(  f(\mathbf{x})  - \frac{\sigma^2}{2}  \nabla \log p_t(\mathbf{x})  \right) \right]
\end{align}
From this formulation one can apply the divergence theorem and obtain the deterministic dynamics similar to [5]. The explanation used in the paper uses the Wasserstein gradient flow structure.

- Poorly written and limited conclusion without discussing the limitations and possible outcomes of the provided guarantees.

Minor:

- The notation of the stationary distribution of the overdamped Langevin dynamics is rather unfortunate, especially considering that then it is used in the same sentences/equations with the parameter $\gamma$ indicating the damping of the underdamped process.
I recommend adapting the notation to avoid confusing the readers unnecessarily.

**Questions:**

- How does the proposed approach relate to the flow matching methods, i.e. [8]?

- How does the method compare with concurrent approaches [7] , [9] and [10]?

**Limitations:**

- The authors do not provide numerical experiments to support their claims.

- They do not provide comparisons with already proposed bounds on either stochastic or deterministic implementations of diffusion based generative models.

- They do not discuss limitations of their approach.




---
**References:**


[1] Chen, Sitan, et al. "Sampling is as easy as learning the score: theory for diffusion models with minimal data assumptions." arXiv preprint arXiv:2209.11215 (2022).

[2] Lee, Holden, Jianfeng Lu, and Yixin Tan. "Convergence of score-based generative modeling for general data distributions." International Conference on Algorithmic Learning Theory. PMLR, 2023.

[3] Maoutsa, D., Reich, S., & Opper, M. (2020). Interacting particle solutions of Fokker–Planck equations through gradient–log–density estimation. Entropy, 22(8), 802.

[4] Song, Y., Sohl-Dickstein, J., Kingma, D. P., Kumar, A., Ermon, S., & Poole, B. (2020). Score-based generative modeling through stochastic differential equations. arXiv preprint arXiv:2011.13456.

[5] Hermoso, A., Homar, V., & Yano, J. I. (2020). Exploring the limits of ensemble forecasting via solutions of the Liouville equation for realistic geophysical models. Atmospheric Research, 246, 105127.

[6] Albergo, Michael S., and Eric Vanden-Eijnden. "Building normalizing flows with stochastic interpolants." arXiv preprint arXiv:2209.15571 (2022).

[7]  Li, Gen, et al. "Towards Faster Non-Asymptotic Convergence for Diffusion-Based Generative Models." arXiv preprint arXiv:2306.09251 (2023).

[8] Benton, Joe, George Deligiannidis, and Arnaud Doucet. "Error Bounds for Flow Matching Methods." arXiv preprint arXiv:2305.16860 (2023).

[9] Lee, Holden, Jianfeng Lu, and Yixin Tan. "Convergence of score-based generative modeling for general data distributions." International Conference on Algorithmic Learning Theory. PMLR, 2023.

[10] Albergo, Michael S., Nicholas M. Boffi, and Eric Vanden-Eijnden. "Stochastic interpolants: A unifying framework for flows and diffusions." arXiv preprint arXiv:2303.08797 (2023).

[9] Jain, Ajay, and Ben Poole. "Journey to the BAOAB-limit: finding effective MCMC samplers for score-based models." NeurIPS 2022 Workshop on Score-Based Methods. 2022.

---

> ### Author Rebuttal · Authors · 2023-08-08
>
> Thank you for the detailed review.
>
> **They propose two variants of the algorithm, one relying on overdamped Langevin dynamics with dimension dependence that scales as $O(d)$, and one with underdamped Langevin dynamics that scales more favourably as $O(\sqrt{d})$ and thus has improved dimension dependence compared to previous approaches.**
>
> To be precise, we did not propose the overdamped version DPOM. As mentioned in the paper, DPOM was proposed **and experimented** in [Son+21a, Section 4.2].
>
> **However I find that the article could have been more accessible for the broader audience of NeurIPS**
>
> Thanks for the feedback. We hope that the main message of our results, that you summarized in the beginning of your review, is accessible to the community.
>
> Besides, we agree that the theory of diffusion models is a difficult topic. We have tried to make the ideas of our proof accessible by writing an outline in Section 4. Once again, the literature on diffusion models is mainly applied, and we believe that this nicely complements the literature.
>
>
> **The work is purely theoretical and the authors do not provide numerical evidence for their claims excluding a simple experiment in the supplement. [...]**
>
> Please refer to our general message for all reviewers.
>
> The experiment we provided is purely illustrative, that’s why we chose to put it in the appendix, and we might remove them from the next version of the paper if it confuses the reader about the theoretical nature of our paper.
>
> **The main theorem of the paper is formulated considering only linear dynamics (Ornstein-Uhlenbeck process) [...]**
>
> Thanks for the suggestion. We will apply the recommended change in our revision of the paper. Please consider this weakness fixed.
>
> To be precise, we work with the Ornstein-Uhlenbeck (OU) process for many reasons: the analysis is involved enough already; our theoretical results are the first ones for this setting and therefore we aim to prove the simplest possible results rather than covering all possible extensions; and from our understanding, the OU setting is already interesting enough for practice.
>
> **The derivation of the probability flow ODE in Eq. 3 is a bit odd, and happens to be correct here because the considered dynamics is conservative [...]**
>
> The Prob flow ODE is the name given in the ML community when they rediscovered this ODE in 2020. However, this ODE is actually an old topic, already considered in optimal transport, see [AGS08]. We preferred to stick to the original approach of [AGS08] for interpretability. The approach you propose is valid, but we do not think that the approach we use is a weakness of the paper.
>
> Besides, for us, the term "conservative" refers to dynamics that preserve the global energy. As a gradient flow, the Prob flow ODE "maximizes" the dissipation of energy, so how can it be conservative?
>
> **Poorly written and limited conclusion without discussing the limitations and possible outcomes of the provided guarantees.**
>
> We will rewrite the conclusion and in particular add the following limitations:
>
> - Our theory only covers the ODE version of the Ornstein–Uhlenbeck process. We leave more general processes for future work.
> - The guarantees we obtain require to learn the score with $L^2$-accuracy $O(\epsilon / \sqrt{L})$, which is *a bit* more stringent than the requirement obtained for DDPM (SDE based diffusion model) in [CLL22, Che+23] (They require $O(\epsilon)$).
> - No large scale experiments. To be clear, we agree that it is a limitation, but we disagree that it is necessary for publication in NeurIPS given our theoretical contribution: we obtain $O(\sqrt{d})$ whereas all prior works, including DPOM, obtain at least $O(d)$.
>
> **MINOR**: Thanks for the suggestion, please consider it fixed.
>
> **Flow matching [8].** This paper is very interesting and relevant to ours. We will add it to our literature review. Note however that **it appeared on arXiv after NeurIPS deadline.** We can compare our paper to [8] along several axes, we choose to mention some of them here:
>
> 1. [8] is more general than our work and therefore the bounds obtained are less precise than ours in the case of the Prob Flow ODE.
>
> In particular:
>
> 2. They cover the Prob flow ODE (without corrector).
> 3. They do not consider discretization in time. More precisely, the discretization error is hidden by their Assumption 1.
> 4. Their result scales exponentially in the parameter $\lambda$ which could depend on the dimension.
>
> TL;DR: [8] is more general than us but less precise in the particular case of the Prob Flow ODE.
>
> **Comparison to [7]**. These are interesting results. Again, **this paper appeared online after NeurIPS deadline**. Using a modified ODE they managed to obtain an improved dependence on the accuracy ($O(1/\sqrt{\epsilon})$ instead of $O(1/\epsilon)$ for us).
> However a major concern we have is that they assume zero score estimation error (see their Assumption 1), whereas we allow for a score estimation error.
>
> **Comparison to [9]**. Already done. See end of Page 9 (starting line 220)
>
> **Comparison to [10]**. This is also a very interesting paper and we will add it to our literature review. It is a dynamical systems paper (rather than a convergence paper like ours) that obtains a number of ODEs and SDEs for generative modeling using a general framework.
>
>
> **They do not provide comparisons with already proposed bounds on either stochastic or deterministic implementations of diffusion based generative models.**
>
> Comparison already done, **with the references that were available before NeurIPS deadline**. Please see the paragraph between Th2 and Th3.
>
>
>
> **We kindly ask you to evaluate our paper with the lens of a theoretician, i.e., in light of our actual contributions (towards theoretical understanding of diffusion models). If you believe that our theoretical contribution is enough for publication please consider raising your score.**

---

> > ### Author Response · Authors · 2023-08-18
> > **Simulations in higher dimension**
> >
> > An update on our simulations:
> >
> > Our paper contains simulations of 500 particles in dimension 5 along 300 iterations. We have modified the code to run simulations in higher dimension (e.g., d = 500).
> >
> >
> > In particular, we observe that in dimension 180, our algorithm's output at iteration 1800 is equivalent to the iteration 300 in dimension 5. In other words, by multiplying the dimension by 36, the algorithm is 6 times slower. Since $6 = \sqrt{36}$, this simulation suggests the scaling $O(\sqrt{d})$ for DPUM, which is mathematically proven in the paper.
> >
> > **We hope that this answers your concern on the toy experiments.**

---

### Author Rebuttal · Authors · 2023-08-07

**TO ALL REVIEWERS and AC:**

We thank all of the reviewers for their hard work in reviewing our article. All reviewers are in agreement regarding the high quality and novelty of our theoretical results, and that our results are of interest to the NeurIPS community. In particular, we leverage recent techniques to accelerate sampling with the Probability Flow ODE: **when using the right corrector, we obtain a complexity that scales as $O(\sqrt{d})$ whereas all other existing complexity results in the field of diffusion models scale at least as $O(d)$.** This better scaling with the dimension specifically matters when generating large scale objects like videos. We believe this is a technical achievement because accelerating sampling is a challenging topic.



However, the reviewers have also discussed our lack of extensive experiments.

*In the field of diffusion models, in recent years there have been tons of "applied" papers claiming experimental breakthroughs published at NeurIPS. Yet, there have been perhaps only a dozen of "theory" papers providing scientific understanding. Moreover, up to one or two exceptions, these applied papers do not provide extensive theory, and we do not think that they need to.*

*So why should we expect a theory paper like ours to provide extensive experiments in order to be published?*

**From reading the reviewers’ opinions, it would seem that our paper clearly passes the bar for a theoretical contribution, and we ask that our paper be evaluated in this regard.**




Besides, our mathematical analysis suggests that the novel corrector we propose could be potentially useful in practice. However, our aim is not to showcase the superiority of our methods via comprehensive experiments. Again, there are thousands of papers purporting to do exactly this, and it is best left for future work. What separates our work from others in this field is to uncover a **mathematical understanding of diffusion models.**

---

> ### Comment · Reviewer_L7v8 · 2023-08-11
> **About the experiments**
>
> Please stop saying the reviewers are requiring _**extensive**_ experiments! First, evaluating your sampler on pre-trained diffusion models **is not difficult** at all and there are no extra training costs. Besides, there are so many available pre-trained diffusion models on different datasets you can use, such as on CIFAR10, FFHQ, LSUN-Bedroom, CelebA, etc. These models are not too big and I believe it is very convenient to use them to evaluate your method.
>
> I think just a simple experiment on **any of the above datasets** would be enough for a theoretical paper like yours. However, only a toy example is provided in the supplementary, which makes it questionable whether the proposed method is effective.
>
> I highly encourage you to add an experiment using just one model I mentioned above and compare your method to existing methods.

---

> > ### Comment · Reviewer_tQnV · 2023-08-12
> >
> > I can understand why reviewers are pushing for more experiments, especially when the authors claimed their algorithms as one of the main contribution. This can potentially be misleading. I would recommend the authors to clarify their message in the paper.
> >
> > The experiments suggested by L7v8 will clearly be a nice addition to this paper (as mentioned in my first round comments as well).
> > However, I'd like to argue for the authors that adding more experiments or the existence of ANY empirical result is not central to the contribution of this specific work. In fact, I would suggest the authors to just remove the toy experiments; these illustrations do not seem to be representative enough. The key contribution of this paper is theoretical in nature and the evaluation of this work should purely be based on whether the results or proof techniques are mathematically interesting (or maybe the clarity of the writing as well). To me, this work clearly improved existing results and provided a fairly novel techinique of achieving polynomial TV convergence, with a high quality writing as well. (Please see my comments and discussions with the authors for justifications of the above points; but I also respect that if other reviewers hold different opinions.)
> >
> >
> > As for the expectation of empirical verifications in theory papers in this domain, I see it as a philosophical question instead of a strictly scientific one. While the community rightly values methodologies that show promising empirical outcomes, works that make attempts to understand and interpret of existing methods should also be appreciated -- even if even if these theoretical probes sometimes operate in constrained or potentially impractical settings. Supporting and encouraging these ``impractical'' but rigorous scientific investigations are crucial to fostering a healthy and diversed research community.

---

> > > ### Comment · Reviewer_Q98a · 2023-08-12
> > > **experiments**
> > >
> > > I basically second tQnV's proposal.  Including real-world simulations like CIFAR10, FFHQ, LSUN-Bedroom, or CelebA may not be needed for this paper.

---

### Decision · Program_Chairs · 2023-09-21

**Decision:**

Accept (poster)

**Comment:**

In this paper, the authors studied the convergence properties of an important class of score-based generative modeling:  (some variations of) the probabilistic flow ODE based implementation. When properly modified by a correction step, the authors established theoretical convergence bounds for the algorithms, and demonstrated improved dimension dependency for smooth data distributions when underdamped Langevin diffusion is incorporated into the probability flow ODE implementation. The paper provides solid theoretical contributions to demonstrate the potential of acceleration via proper correction. As the reviewers pointed out, the paper would have benefited from enhanced numerical experiments, given that the algorithms proposed and analyzed in this paper differ from the implementations widely used in practice. In addition, since the schemes analyzed in this paper are not exactly probability flow ODE (due to the correction steps), the current title and introduction might be a little misleading.